# Provably expressive temporal graph networks

**Amauri H. Souza**[1], **Diego Mesquita**[2], **Samuel Kaski**[1,3], **Vikas Garg**[1,4]

[1]Aalto University  [2]Getulio Vargas Foundation  [3]University of Manchester  [4]YaiYai Ltd

{amauri.souza, samuel.kaski}@aalto.fi, diego.mesquita@fgv.br, vgarg@csail.mit.edu

## Abstract

Temporal graph networks (TGNs) have gained prominence as models for embedding dynamic interactions, but little is known about their theoretical underpinnings. We establish fundamental results about the representational power and limits of the two main categories of TGNs: those that aggregate temporal walks (WA-TGNs), and those that augment local message passing with recurrent memory modules (MP-TGNs). Specifically, novel constructions reveal the inadequacy of MP-TGNs and WA-TGNs, proving that neither category subsumes the other. We extend the 1-WL (Weisfeiler-Leman) test to temporal graphs, and show that the most powerful MP-TGNs should use injective updates, as in this case they become as expressive as the temporal WL. Also, we show that sufficiently deep MP-TGNs cannot benefit from memory, and MP/WA-TGNs fail to compute graph properties such as girth.

These theoretical insights lead us to PINT — a novel architecture that leverages injective temporal message passing and relative positional features. Importantly, PINT is provably more expressive than both MP-TGNs and WA-TGNs. PINT significantly outperforms existing TGNs on several real-world benchmarks.

## 1  Introduction

Graph neural networks (GNNs) [11, 30, 36, 39] have recently led to breakthroughs in many applications [7, 28, 31] by resorting to message passing between neighboring nodes in input graphs. While message passing imposes an important inductive bias, it does not account for the dynamic nature of interactions in time-evolving graphs arising from many real-world domains such as social networks and bioinformatics [16, 40]. In several scenarios, these temporal graphs are only given as a sequence of timestamped events. Recently, temporal graph nets (TGNs) [16, 27, 32, 38, 42] have emerged as a prominent learning framework for temporal graphs and have become particularly popular due to their outstanding predictive performance. Aiming at capturing meaningful structural and temporal patterns, TGNs combine a variety of building blocks, such as self-attention [33, 34], time encoders [15, 41], recurrent models [5, 13], and message passing [10].

Unraveling the learning capabilities of (temporal) graph networks is imperative to understanding their strengths and pitfalls, and designing better, more nuanced models that are both theoretically well-grounded and practically efficacious. For instance, the enhanced expressivity of higher-order GNNs has roots in the inadequacy of standard message-passing GNNs to separate graphs that are indistinguishable by the Weisfeiler-Leman isomorphism test, known as 1-WL test or color refinement algorithm [21, 22, 29, 37, 43]. Similarly, many other notable advances on GNNs were made possible by untangling their ability to generalize [9, 17, 35], extrapolate [45], compute graph properties [4, 6, 9], and express Boolean classifiers [1]; by uncovering their connections to distributed algorithms [19, 29], graph kernels [8], dynamic programming [44], diffusion processes [3], graphical models [46], and combinatorial optimization [2]; and by analyzing their discriminative power [20, 23]. In stark contrast, the theoretical foundations of TGNs remain largely unexplored. For instance, unresolved questions include: How does the expressive power of existing TGNs compare? When do TGNs fail? Can we improve the expressiveness of TGNs? What are the limits on the power of TGNs?

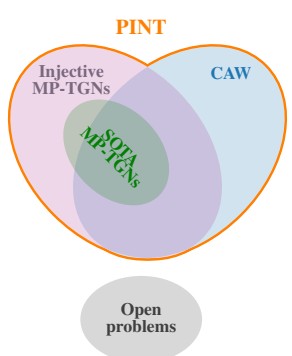

| Overview of the theoretical results | |
|---|---|
| Relationship between CTDGs and DTDGs | Prop. 1 |
| Injective MP-TGNs $\succeq$ MP-TGNs | Prop. 2 |
| Sufficiently deep MP-TGNs do not need memory | Prop. 3 |
| SOTA MP-TGNs $\prec$ Injective MP-TGNs | Prop. 4 |
| MP-TGNs $\not\succ$ WA-TGNs and WA-TGNs $\not\succ$ MP-TGNs | Prop. 5 |
| Injective MP-TGNs $\cong$ temporal-WL test | Prop. 6 |
| MP-TGNs/CAWs cannot recognize graph properties | Prop. 7 |
| Constructing injective temporal MP | Prop. 8 |
| **PINT** (ours) $\succ$ both MP-TGNs and WA-TGNs | Prop. 9 |
| Limitations of PINT | Prop. 10 |

Figure 1: Schematic diagram and summary of our contributions.

We establish a series of results to address these fundamental questions. We begin by showing that discrete-time dynamic graphs (DTDGs) can always be converted to continuous-time analogues (CTDGs) without loss of information, so we can focus on analyzing the ability of TGNs to distinguish nodes/links of CTDGs. We consider a general framework for message-passing TGNs (MP-TGNs) [27] that subsumes a wide variety of methods [e.g., 16, 32, 42]. We prove that equipping MP-TGNs with injective aggregation and update functions leads to the class of most expressive *anonymous MP-TGNs* (i.e., those that do not leverage node ids). Extending the color-refinement algorithm to temporal settings, we show that these most powerful MP-TGNs are as expressive as the temporal WL method. Notably, existing MP-TGNs do not enforce injectivity. We also delineate the role of memory in MP-TGNs: nodes in a network with only a few layers of message passing fail to aggregate information from a sufficiently wide receptive field (i.e., from distant nodes), so memory serves to offset this highly local view with additional global information. In contrast, sufficiently deep architectures obviate the need for memory modules.

Different from MP-TGNs, walk-aggregating TGNs (WA-TGNs) such as CAW [38] obtain representations from anonymized temporal walks. We provide constructions that expose shortcomings of each framework, establishing that WA-TGNs can distinguish links in cases where MP-TGNs fail and vice-versa. Consequently, neither class is more expressive than the other. Additionally, we show that MP-TGNs and CAWs cannot decide temporal graph properties such as diameter, girth, or number of cycles. Strikingly, our analysis unravels the subtle relationship between the walk computations in CAWs and the MP steps in MP-TGNs.

Equipped with these theoretical insights, we propose PINT (short for *position-encoding injective temporal graph net*), founded on a new temporal layer that leverages the strengths of both MP-TGNs and WA-TGNs. Like the most expressive MP-TGNs, PINT defines injective message passing and update steps. PINT also augments memory states with novel relative positional features, and these features can replicate all the discriminative benefits available to WA-TGNs. Interestingly, the time complexity of computing our positional features is less severe than the sampling overhead in CAW, thus PINT can often be trained faster than CAW. Importantly, we establish that PINT is provably more expressive than CAW as well as MP-TGNs.

**Our contributions** are three-fold:

- a rigorous theoretical foundation for TGNs is laid - elucidating the role of memory, benefits of injective message passing, limits of existing TGN models, temporal extension of the 1-WL test and its implications, impossibility results about temporal graph properties, and the relationship between main classes of TGNs — as summarized in Figure 1;

- explicit injective temporal functions are introduced, and a novel method for temporal graphs is proposed that is provably more expressive than state-of-the-art TGNs;

- extensive empirical investigations underscore practical benefits of this work. The proposed method is either competitive or significantly better than existing models on several real benchmarks for dynamic link prediction, in transductive as well as inductive settings.

## 2 Preliminaries

We denote a *static graph* $G$ as a tuple $(V, E, \mathcal{X}, \mathcal{E})$, where $V = \{1, 2, \ldots, n\}$ denotes the set of nodes and $E \subseteq V \times V$ the set of edges. Each node $u \in V$ has a feature vector $x_u \in \mathcal{X}$ and each edge $(u, v) \in E$ has a feature vector $e_{uv} \in \mathcal{E}$, where $\mathcal{X}$ and $\mathcal{E}$ are countable sets of features.

**Dynamic graphs** can be roughly split according to their discrete- or continuous-time nature [14]. A *discrete-time dynamic graph* (DTDG) is of a sequence of graph snapshots $(G_1, G_2, \ldots)$ usually sampled at regular intervals, each snapshot being a static graph $G_t = (V_t, E_t, \mathcal{X}_t, \mathcal{E}_t)$.

A *continuous-time dynamic graph* (CTDG) evolves with node- and edge-level *events*, such as addition and deletion. We represent a CTDG as a sequence of time-stamped multi-graphs $(\mathsf{G}(t_0), \mathsf{G}(t_1), \ldots)$ such that $t_k < t_{k+1}$, and $\mathsf{G}(t_{k+1})$ results from updating $\mathsf{G}(t_k)$ with all events at time $t_{k+1}$. We assume no event occurs between $t_k$ and $t_{k+1}$. We denote an interaction (i.e., edge addition event) between nodes $u$ and $v$ at time $t$ as a tuple $(u, v, t)$ associated with a feature vector $e_{uv}(t)$. Unless otherwise stated, interactions correspond to undirected edges, i.e., $(u, v, t)$ is a shorthand for $(\{u, v\}, t)$.

Noting that CTDGs allow for finer (irregular) temporal resolution, we now formalize the intuition that DTDGs can be reduced to and thus analyzed as CTDGs, but the converse may need extra assumptions.

**Proposition 1** (Relationship between DTDG and CTDG). *For any DTDG we can build a CTDG with the same sets of node and edge features that contains the same information, i.e., we can reconstruct the original DTDG from the converted CTDG. The converse holds if the CTDG timestamps form a subset of a uniformly spaced countable set.*

Following the usual practice [16, 38, 42], we focus on CTDGs with edge addition events (see Appendix E for a discussion on deletion). Thus, we can represent temporal graphs as sets $\mathcal{G}(t) = \{(u_k, v_k, t_k) \mid t_k < t\}$. We also assume each distinct node $v$ in $\mathcal{G}(t)$ has an initial feature vector $x_v$.

**Message-passing temporal graph nets (MP-TGNs).** Rossi et al. [27] introduced MP-TGN as a general representation learning framework for temporal graphs. The goal is to encode the graph dynamics into node embeddings, capturing information that is relevant for the task at hand. To achieve this, MP-TGNs rely on three main ingredients: memory, aggregation, and update. Memory comprises a set of vectors that summarizes the history of each node, and is updated using a recurrent model whenever an event occurs. The aggregation and update components resemble those in message-passing GNNs, where the embedding of each node is refined using messages from its neighbors.

We define the *temporal neighborhood* of node $v$ at time $t$ as $\mathcal{N}(v, t) = \{(u, e_{uv}(t'), t') \mid \exists (u, v, t') \in \mathcal{G}(t)\}$, i.e., the set of neighbor/feature/timestamp triplets from all interactions of node $v$ prior to $t$. MP-TGNs compute the temporal representation $h_v^{(\ell)}(t)$ of $v$ at layer $\ell$ by recursively applying

$$\tilde{h}_v^{(\ell)}(t) = \text{AGG}^{(\ell)}(\{\!\{(h_u^{(\ell-1)}(t), t - t', e) \mid (u, e, t') \in \mathcal{N}(v, t)\}\!\}) \tag{1}$$

$$h_v^{(\ell)}(t) = \text{UPDATE}^{(\ell)}\left(h_v^{(\ell-1)}(t), \tilde{h}_v^{(\ell)}(t)\right), \tag{2}$$

where $\{\!\{\cdot\}\!\}$ denotes multisets, $h_v^{(0)}(t) = s_v(t)$ is the *state* of $v$ at time $t$, and $\text{AGG}^{(\ell)}$ and $\text{UPDATE}^{(\ell)}$ are arbitrary parameterized functions. The memory block updates the states as events occur. Let $\mathcal{J}(v, t)$ be the set of events involving $v$ at time $t$. The state of $v$ is updated due to $\mathcal{J}(v, t)$ as

$$m_v(t) = \text{MEMAGG}(\{\!\{[s_v(t), s_u(t), t - t_v, e_{vu}(t)] \mid (v, u, t) \in \mathcal{J}(v, t)\}\!\}) \tag{3}$$

$$s_v(t^+) = \text{MEMUPDATE}(s_v(t), m_v(t)), \tag{4}$$

where $s_v(0) = x_v$ (initial node features), $s_v(t^+)$ denotes the updated state of $v$ due to events at time $t$, and $t_v$ denotes the time of the last update to $v$. MEMAGG combines information from simultaneous events involving node $v$ and MEMUPDATE usually implements a gated recurrent unit (GRU) [5]. Notably, some MP-TGNs do not use memory, or equivalently, they employ *identity memory*, i.e., $s_v(t) = x_v$ for all $t$. We refer to Appendix A for further details.

**Causal Anonymous Walks (CAWs).** Wang et al. [38] proposed CAW as an approach for link prediction on temporal graphs. To predict if an event $(u, v, t)$ occurs, CAW first obtains sets $S_u$ and $S_v$ of temporal walks starting at nodes $u$ and $v$ at time $t$. An $(L - 1)$-length *temporal walk* is represented as $W = ((w_1, t_1), (w_2, t_2), \ldots, (w_L, t_L))$, with $t_1 > t_2 > \cdots > t_L$ and $(w_{i-1}, w_i, t_i) \in \mathcal{G}(t)$ $\forall i > 1$. Note that when predicting $(u, v, t)$, we have walks starting at time $t_1 = t$. Then, CAW anonymizes walks replacing each node $w$ with a set $I_{\text{CAW}}(w; S_u, S_v) = \{g(w; S_u), g(w; S_v)\}$ of two

feature vectors. The $\ell$-th entry of $g(w; S_u)$ stores how many times $w$ appears at the $\ell$-th position in a walk of $S_u$, i.e. $g(w, S_u)[\ell] = |\{W \in S_u : (w, t_\ell) = W_\ell\}|$ where $W_\ell$ is $\ell$-th pair of $W$.

To encode a walk $W$ with respect to the sets $S_u$ and $S_v$, CAW applies $\text{ENC}(W; S_u, S_v) = \text{RNN}([f_1(I_{\text{CAW}}(w_i; S_u, S_v))\|f_2(t_{i-1} - t_i)]_{i=1}^L)$ where $f_1$ is a permutation-invariant function, $f_2$ is a time encoder, and $t_0 = t_1 = t$. Finally, CAW combines the embeddings of each walk in $S_u \cup S_v$ using mean-pooling or self-attention to obtain the representation for the event $(u, v, t)$.

In practice, TGNs often rely on sampling schemes for computational reasons. However we are concerned with the expressiveness of TGNs, so our analysis assumes complete structural information, i.e., $S_u$ is the set of all temporal walks from $u$ and MP-TGNs combine information from all neighbors.

## 3 The representational power and limits of TGNs

We now study the expressiveness of TGNs on node/edge-level prediction. We also establish connections to a variant of the WL test and show limits of specific TGN models. Proofs are in Appendix B.

### 3.1 Distinguishing nodes with MP-TGNs

We analyze MP-TGNs w.r.t. their ability to map different nodes to different locations in the embedding space. In particular, we say that an $L$-layer MP-TGN distinguishes two nodes $u, v$ of a temporal graph at time $t$, if the last layer embeddings of $u$ and $v$ are different, i.e., $h_u^{(L)}(t) \neq h_v^{(L)}(t)$.

We can describe the MP computations of a node $v$ at time $t$ via its *temporal computation tree* (TCT) $T_v(t)$. $T_v(t)$ has $v$ as its root and height equal to the number of MP-TGN layers $L$. We will keep the dependence on depth $L$ implicit for notational simplicity. For each element $(u, e, t') \in \mathcal{N}(v, t)$ associated with $v$, we have a node, say $i$, in the next layer of the TCT linked to the root by an edge annotated with $(e, t')$. The remaining TCT layers are built recursively using the same mechanism. We denote by $\sharp_v^t$ the (possibly many-to-one) operator that maps nodes in $T_v(t)$ back to nodes in $\mathcal{G}(t)$, e.g., $\sharp_v^t i = u$. Each node $i$ in $T_v(t)$ has a state vector $s_i = s_{\sharp_v^t i}(t)$. To get the embedding of the root $v$, information is propagated bottom-up, i.e., starting from the leaves all the way up to the root — each node aggregates the message from the layer below and updates its representation along the way. Whenever clear from context, we denote $\sharp_v^t$ simply as $\sharp$ for a cleaner notation.

We study the expressive power of MP-TGNs through the lens of functions on multisets adapted to temporal settings, i.e., comprising triplets of node states, edge features, and timestamps. Intuitively, injective functions 'preserve' the information as it is propagated, so should be essential for maximally expressive MP-TGNs. We formalize this idea in Lemma 1 and Proposition 2 via Definition 1.

**Definition 1** (Isomorphic TCTs). *Two TCTs $T_z(t)$ and $T_{z'}(t)$ at time $t$ are isomorphic if there is a bijection $f : V(T_z(t)) \to V(T_{z'}(t))$ between the nodes of the trees such that the following holds:*

$$(u, v, t') \in E(T_z(t)) \iff (f(u), f(v), t') \in E(T_{z'}(t))$$

$$\forall(u, v, t') \in E(T_z(t)) : e_{uv}(t') = e_{f(u)f(v)}(t') \text{ and } \forall u \in V(T_z(t)) : s_u = s_{f(u)} \text{ and } k_u = k_{f(u)}$$

*Here, $k_u$ denotes the level (depth) of node $u$ in the tree. The root node has level $0$, and for a node $u$ with level $k_u$, the children of $u$ have level $k_u + 1$.*

**Lemma 1.** *If an MP-TGN $Q$ with $L$ layers distinguishes two nodes $u, v$ of a dynamic graph $\mathcal{G}(t)$, then the $L$-depth TCTs $T_u(t)$ and $T_v(t)$ are not isomorphic.*

For non-isomorphic TCTs, Proposition 2 shows that improving MP-TGNs with *injective* message passing layers suffices to achieve node distinguishability, extending results from static GNNs [43].

**Proposition 2** (MP-TGNs with injective message passing). *If the $L$-depth TCTs of two nodes $u, v$ of a temporal graph $\mathcal{G}(t)$ at time $t$ are not isomorphic, then an MP-TGN $Q$ with $L$ layers and injective aggregation and update functions at each layer is able to distinguish nodes $u$ and $v$.*

So far, we have considered TCTs with general memory modules, i.e., nodes are annotated with memory states. However, an important question remains: *How does the expressive power of MP-TGNs change as a function of the memory?* Naturally, different implementation of memory modules can determine whether the TCTs of two nodes are isomorphic or not. Thus, most expressive MP-TGNs should also employ injective memory aggregation/update. However, Proposition 3 shows that adding GRU-based memory does not increase the expressiveness of suitably deep MP-TGNs.

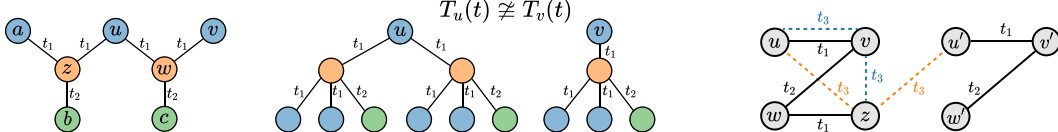

Figure 2: **Limitations of TGNs**. [*Left*] Temporal graph with nodes $u, v$ that TGN-Att/TGAT cannot distinguish. Colors are node features, edge features are identical, and $t_3 > t_2 > t_1$. [*Center*] TCTs of $u$ and $v$ are non-isomorphic. However, the attention layers of TGAT/TGN-Att compute weighted averages over a same multiset of values, returning identical messages for $u$ and $v$. [*Right*] MP-TGNs fail to distinguish the events $(u, v, t_3)$ and $(v, z, t_3)$ as TCTs of $z$ and $u$ are isomorphic. Meanwhile, CAW cannot separate $(u, z, t_3)$ and $(u', z, t_3)$: the 3-depth TCTs of $u$ and $u'$ are not isomorphic, but the temporal walks from $u$ and $u'$ have length 1, keeping CAW from capturing structural differences.

**Proposition 3** (The role of memory). *Let $\mathcal{Q}_L^{[M]}$ denote the class of MP-TGNs with recurrent memory and $L$ layers. Similarly, we denote by $\mathcal{Q}_L$ the family of memoryless MP-TGNs with $L$ layers. Let $\Delta$ be the temporal diameter of $\mathcal{G}(t)$ (see Definition B2). Then, it holds that:*

1. *If $L < \Delta$: $\mathcal{Q}_L^{[M]}$ is strictly more powerful than $\mathcal{Q}_L$ in distinguishing nodes of $\mathcal{G}(t)$;*
2. *For any $L$: $\mathcal{Q}_{L+\Delta}$ is at least as powerful as $\mathcal{Q}_L^{[M]}$ in distinguishing nodes of $\mathcal{G}(t)$.*

The MP-TGN framework is rather general and subsumes many modern methods for temporal graphs [e.g., 16, 32, 42]. We now analyze the theoretical limitations of two concrete instances of MP-TGNs: TGAT [42] and TGN-Att [27]. Remarkably, these models are among the best-performing MP-TGNs. Nonetheless, we can show that there are nodes of very simple temporal graphs that TGAT and TGN-Att cannot distinguish (see Figure 2). We formalize this in Proposition 4 by establishing that there are cases in which TGNs with injective layers can succeed, but TGAT and TGN-Att cannot.

**Proposition 4** (Limitations of TGAT/TGN-Att). *There exist temporal graphs containing nodes $u, v$ that have non-isomorphic TCTs, yet no TGAT nor TGN-Att with mean memory aggregator (i.e., using MEAN as MEMAGG) can distinguish $u$ and $v$.*

This limitation stems from the fact that the attention mechanism employed by TGAT and TGN-Att is proportion invariant [26]. The memory module of TGN-Att cannot counteract this limitation due to its mean-based aggregation scheme. We provide more details in Appendix B.6.

### 3.2 Predicting temporal links

Models for dynamic graphs are usually trained and evaluated on temporal link prediction [18], which consists in predicting whether an event would occur at a given time. To predict an event between nodes $u$ and $v$ at $t$, MP-TGNs combine the node embeddings $h_u^{(L)}(t)$ and $h_v^{(L)}(t)$, and push the resulting vector through an MLP. On the other hand, CAW is originally designed for link prediction tasks and directly computes edge embeddings, bypasssing the computation of node representations.

We can extend the notion of node distinguishability to edges/events. We say that a model distinguishes two synchronous events $\gamma = (u, v, t)$ and $\gamma' = (u', v', t)$ of a temporal graph if it assigns different edge embeddings $h_\gamma \neq h_{\gamma'}$ for $\gamma$ and $\gamma'$. Proposition 5 asserts that CAWs are not strictly more expressive than MP-TGNs, and vice-versa. Intuitively, CAW's advantage over MP-TGNs lies in its ability to exploit node identities and capture correlation between walks. However, CAW imposes temporal constraints on random walks, i.e., walks have timestamps in decreasing order, which can limit its ability to distinguish events. Figure 2(Right) sketches constructions for Proposition 5.

**Proposition 5** (Limitations of MP-TGNs and CAW). *There exist distinct synchronous events of a temporal graph that CAW can distinguish but MP-TGNs with injective layers cannot, and vice-versa.*

### 3.3 Connections with the WL test

The Weisfeiler-Leman test (1-WL) has been used as a key tool to analyze the expressive power of GNNs. We now study the power of MP-TGNs under a temporally-extended version of 1-WL, and prove negative results regarding whether TGNs can recognize properties of temporal graphs.

**Temporal WL test.** We can extend the WL test for temporal settings in a straightforward manner by exploiting the equivalence between temporal graphs and multi-graphs with timestamped edges [24]. In particular, the temporal variant of 1-WL assigns colors for all nodes in an input dynamic graph $\mathcal{G}(t)$ by applying the following iterative procedure:

*Initialization*: The colors of all nodes in $\mathcal{G}(t)$ are initialized using the initial node features: $\forall v \in V(\mathcal{G}(t)), c^0(v) = x_v$. If node features are not available, all nodes receive identical colors;

*Refinement*: At step $\ell$, the colors of all nodes are refined using a hash (injective) function: for all $v \in V(\mathcal{G}(t))$, we apply $c^{\ell+1}(v) = \text{HASH}(c^\ell(v), \{\!\{(c^\ell(u), e_{uv}(t'), t') : (u, v, t') \in \mathcal{G}(t)\}\!\})$;

*Termination*: The test is carried out for two temporal graphs at time $t$ in parallel and stops when the multisets of corresponding colors diverge, returning non-isomorphic. If the algorithm runs until the number of different colors stops increasing, the test is deemed inconclusive.

We note that the temporal WL test trivially reduces to the standard 1-WL test if all timestamps and edge features are identical. The resemblance between MP-TGNs and GNNs and their corresponding WL tests suggests that the power of MP-TGNs is bounded by the temporal WL test. Proposition 6 conveys that MP-TGNs with injective layers are as powerful as the temporal WL test.

**Proposition 6.** *Assume finite spaces of initial node features $\mathcal{X}$, edge features $\mathcal{E}$, and timestamps $\mathcal{T}$. Let the number of events of any temporal graph be bounded by a fixed constant. Then, there is an MP-TGN with suitable parameters using injective aggregation/update functions that outputs different representations for two temporal graphs if and only if the temporal-WL test outputs 'non-isomorphic'.*

A natural consequence of the limited power of MP-TGNs is that even the most powerful MP-TGNs fail to distinguish relevant graph properties, and the same applies to CAWs (see Proposition 7).

**Proposition 7.** *There exist non-isomorphic temporal graphs differing in properties such as diameter, girth, and total number of cycles, which cannot be differentiated by MP-TGNs and CAWs.*

Figure 3 provides a construction for Proposition 7. The temporal graphs $\mathcal{G}(t)$ and $\mathcal{G}'(t)$ differ in diameter ($\infty$ vs. 3), girth (3 vs. 6), and number of cycles (2 vs. 1). By inspecting the TCTs, one can observe that, for any node in $\mathcal{G}(t)$, there is a corresponding one in $\mathcal{G}'(t)$ whose TCTs are isomorphic, e.g., $T_{u_1}(t) \cong T_{u_1'}(t)$ for $t > t_3$. As a result, the multisets of node embeddings for these temporal graphs are identical. We provide more details and a construction - where CAW fails to decide properties - in the Appendix.

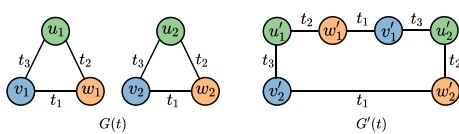

Figure 3: Examples of temporal graphs for which MP-TGNs cannot distinguish the diameter, girth, and number of cycles.

## 4 Position-encoding injective temporal graph net

We now leverage insights from our analysis in Section 3 to build more powerful TGNs. First, we discuss how to build injective aggregation and update functions in the temporal setting. Second, we propose an efficient scheme to compute positional features based on counts from TCTs. In addition, we show that the proposed method, called *position-encoding injective temporal graph net* (PINT), is more powerful than both WA-TGNs and MP-TGNs in distinguishing events in temporal graphs.

**Injective temporal aggregation.** An important design principle in TGNs is to prioritize (give higher importance to) events based on recency [38, 42]. Proposition 8 introduces an injective aggregation scheme that captures this principle using linearly exponential time decay.

**Proposition 8** (Injective function on temporal neighborhood). *Let $\mathcal{X}$ and $\mathcal{E}$ be countable, and $\mathcal{T}$ countable and bounded. There exists a function $f$ and scalars $\alpha$ and $\beta$ such that $\sum_i f(x_i, e_i)\alpha^{-\beta t_i}$ is unique on any multiset $M = \{\!\{(x_i, e_i, t_i)\}\!\} \subseteq \mathcal{X} \times \mathcal{E} \times \mathcal{T}$ with $|M| < N$, where $N$ is a constant.*

Leveraging Proposition 8 and the approximation capabilities of multi-layer perceptrons (MLPs), we propose *position-encoding injective temporal graph net* (PINT). In particular, PINT computes the embedding of node $v$ at time $t$ and layer $\ell$ using the following message passing steps:

$$\tilde{h}_v^{(\ell)}(t) = \sum_{(u,e,t') \in \mathcal{N}(v,t)} \text{MLP}_{\text{agg}}^{(\ell)}\left(h_u^{(\ell-1)}(t) \,\|\, e\right) \alpha^{-\beta(t-t')} \tag{5}$$

$$h_v^{(\ell)}(t) = \text{MLP}_{\text{upd}}^{(\ell)}\left(h_v^{(\ell-1)}(t) \,\|\, \tilde{h}_v^{(\ell)}(t)\right) \tag{6}$$

where $\|$ denotes concatenation, $h_v^{(0)} = s_v(t)$, $\alpha$ and $\beta$ are scalar (hyper-)parameters, and $\text{MLP}_{\text{agg}}^{(\ell)}$ and $\text{MLP}_{\text{upd}}^{(\ell)}$ denote the nonlinear transformations of the aggregation and update steps, respectively.

We note that to guarantee that the MLPs in PINT implement injective aggregation/update, we must further assume that the edge and node features (states) take values in a finite support. In addition,

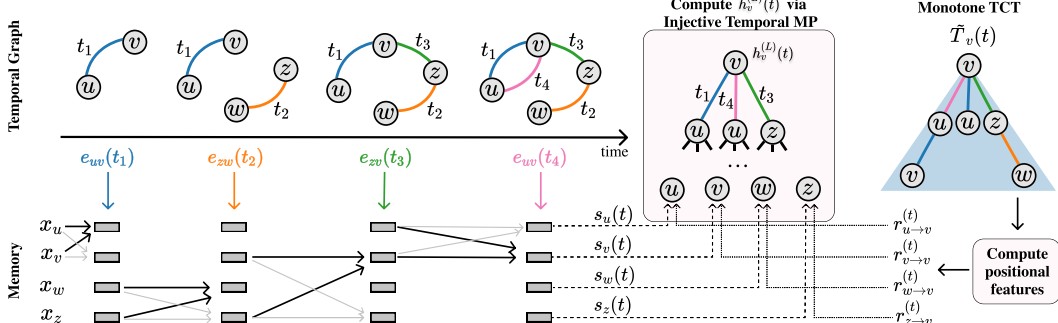

Figure 5: **PINT.** Following the MP-TGN protocol, PINT updates memory states as events unroll. Meanwhile, we use Eqs. (7-11) to update positional features. To extract the embedding for node $v$, we build its TCT, annotate nodes with memory + positional features, and run (injective) MP.

we highlight that there may exist many other ways to achieve injective temporal MP — we have presented a solution that captures the 'recency' inductive bias of real-world temporal networks.

**Relative positional features.** To boost the power of PINT, we propose augmenting memory states with *relative* positional features. These features count how many temporal walks of a given length exist between two nodes, or equivalently, how many times nodes appear at different levels of TCTs.

Formally, let $P$ be the $d \times d$ matrix obtained by padding a $(d-1)$-dimensional identity matrix with zeros on its top row and its rightmost column. Also, let $r_{j \to u}^{(t)} \in \mathbb{N}^d$ denote the positional feature vector of node $j$ relative to $u$'s TCT at time $t$. For each event $(u, v, t)$, with $u$ and $v$ not participating in other events at $t$, we recursively update the positional feature vectors as

$$\mathcal{V}_i^{(0)} = \{i\} \quad \forall i \tag{7}$$
$$\mathcal{V}_u^{(t^+)} = \mathcal{V}_v^{(t^+)} = \mathcal{V}_v^{(t)} \cup \mathcal{V}_u^{(t)} \tag{9}$$

$$r_{i \to j}^{(0)} = \begin{cases} [1, 0, \ldots, 0]^\top & \text{if } i = j \\ [0, 0, \ldots, 0]^\top & \text{if } i \neq j \end{cases} \tag{8}$$
$$r_{i \to v}^{(t^+)} = P \, r_{i \to u}^{(t)} + r_{i \to v}^{(t)} \quad \forall i \in \mathcal{V}_u^{(t)} \tag{10}$$
$$r_{j \to u}^{(t^+)} = P \, r_{j \to v}^{(t)} + r_{j \to u}^{(t)} \quad \forall j \in \mathcal{V}_v^{(t)} \tag{11}$$

where we use $t^+$ to denote values "right after" $t$. The set $\mathcal{V}_i$ keeps track of the nodes for which we need to update positional features when $i$ participates in an interaction. For simplicity, we have assumed that there are no other events involving $u$ or $v$ at time $t$. Appendix B.10 provides equations for the general case where nodes can participate in multiple events at the same timestamp.

The value $r_{i \to v}^{(t)}[k]$ (the $k$-th component of $r_{i \to v}^{(t)}$) corresponds to how many different ways we can get from $v$ to $i$ in $k$ steps through temporal walks. Additionally, we provide in Lemma 2 an interpretation of relative positional features in terms of the so-called monotone TCTs (Definition 2). In this regard, Figure 4 shows how the TCT of $v$ evolves due to an event $(u, v, t)$ and provides an intuition about the updates in Eqs. 10-11. The procedure amounts to appending the monotone TCT of $u$ to the first level of the monotone TCT of $v$.

**Definition 2.** *The monotone TCT of a node $u$ at time $t$, denoted by $\tilde{T}_u(t)$, is the maximal subtree of the TCT of $u$ s.t. for any path $p = (u, t_1, u_1, t_2, u_2, \ldots)$ from the root $u$ to leaf nodes of $\tilde{T}_u(t)$ time monotonically decreases, i.e., we have that $t_1 > t_2 > \cdots$.*

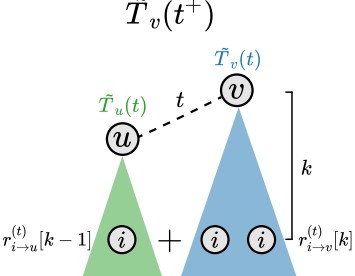

Figure 4: The effect of $(u, v, t)$ on the monotone TCT of $v$. Also, note how the positional features of a node $i$, relative to $v$, can be incrementally updated.

**Lemma 2.** *For any pair of nodes $i, u$ of a temporal graph $\mathcal{G}(t)$, the $k$-th component of the positional feature vector $r_{i \to u}^{(t)}$ stores the number of times $i$ appears at the $k$-th layer of the monotone TCT of $u$.*

**Edge and node embeddings.** To obtain the embedding $h_\gamma$ for an event $\gamma = (u, v, t)$, an $L$-layer PINT computes embeddings for node $u$ and $v$ using $L$ steps of temporal message passing. However, when computing the embedding $h_u^L(t)$ of $u$, we concatenate node states $s_j(t)$ with the positional features $r_{j \to u}^{(t)}$ and $r_{j \to v}^{(t)}$ for all node $j$ in the $L$-hop temporal neighborhood of $u$. We apply the same procedure to obtain $h_v^L(t)$, and then combine $h_v^L(t)$ and $h_u^L(t)$ using a readout function.

Similarly, to compute representations for node-level prediction, for each node $j$ in the $L$-hop neighborhood of $u$, we concatenate node states $s_j(t)$ with features $r_{j \to u}^{(t)}$. Then, we use our injective MP to combine the information stored in $u$ and its neighboring nodes. Figure 5 illustrates the process.

Notably, Proposition 9 states that PINT is strictly more powerful than existing TGNs. In fact, the relative positional features mimic the discriminative power of WA-TGNs, while eliminate their temporal monotonicity constraints. Additionally, PINT can implement injective temporal message passing (either over states or states + positional features), akin to maximally-expressive MP-TGNs.

**Proposition 9** (Expressiveness of PINT: link prediction). *PINT (with relative positional features) is strictly more powerful than MP-TGNs and CAWs in distinguishing events in temporal graphs.*

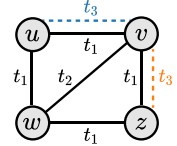

**When does PINT fail?** Naturally, whenever the TCTs (annotated with positional features) for the endpoints of two edges $(u, v, t)$ and $(u', v', t)$ are pairwise isomorphic, PINT returns the same edge embedding and is not able to differentiate the events. Figure 6 shows an example in which this happens — we assume that all node/edge features are identical. Due to graph symmetries, $u$ and $z$ occur the same number of times in each level of $v$'s monotone TCT. Also, the sets of temporal walks starting at $u$ and $z$ are identical if we swap the labels of these nodes. Importantly, CAWs and MP-TGNs also fail here, as stated in Proposition 9.

Figure 6: PINT cannot distinguish the events $(u, v, t_3)$ and $(v, z, t_3)$.

**Proposition 10** (Limitations of PINT). *There are synchronous events of temporal graphs that PINT cannot distinguish (as seen in Figure 6).*

**Implementation and computational cost.** The online updates for PINT's positional features have complexity $\mathcal{O}\left(d\,|\mathcal{V}_u^{(t)}| + d\,|\mathcal{V}_v^{(t)}|\right)$. Similarly to CAW's sampling procedure, our online update is a sequential process better done in CPUs. However, while CAW may require significant CPU-GPU memory exchange — proportional to both the number of walks and their depth —, we only communicate the positional features. We can also speed-up the training of PINT by pre-computing the positional features for each batch, avoiding redundant computations at each epoch. Apart from positional features, the computational cost of PINT is similar to that of TGN-Att. Following standard MP-TGN procedure, we control the branching factor of TCTs using neighborhood sampling.

Note that the positional features monotonically increase with time, which is undesirable for practical generalization purposes. Since our theoretical results hold for any fixed $t$, this issue can be solved by dividing the positional features by a time-dependent normalization factor. Nonetheless, we have found that employing $L_1$-normalization leads to good empirical results for all evaluated datasets.

## 5 Experiments

We now assess the performance of PINT on several popular and large-scale benchmarks for TGNs. We run experiments using PyTorch [25] and code is available at `www.github.com/AaltoPML/PINT`.

**Tasks and datasets.** We evaluate PINT on dynamic link prediction, closely following the evaluation setup employed by Rossi et al. [27] and Xu et al. [42]. We use six popular benchmark datasets: Reddit, Wikipedia, Twitter, UCI, Enron, and LastFM [16, 27, 38, 42]. Notably, UCI, Enron, and LastFM are non-attributed networks, i.e., they do not contain feature vectors associated with the events. Node features are absent in all datasets, thus following previous works we set them to vectors of zeros [27, 42]. Since Twitter is not publicly available, we follow the guidelines by Rossi et al. [27] to create our version. We provide more details regarding datasets in the supplementary material.

**Baselines.** We compare PINT against five prominent TGNs: Jodie [16], DyRep [32], TGAT [42], TGN-Att [27], and CAW [38]. For completeness, we also report results using two static GNNs: GAT [34] and GraphSage [12]. Since we adopt the same setup as TGN-Att, we use their table numbers for all baselines but CAW on Wikipedia and Reddit. The remaining results were obtained using the implementations and guidelines available from the official repositories. As an ablation study, we also include a version of PINT without relative positional features in the comparison. We provide detailed information about hyperparameters and the training of each model in the supplementary material.

**Experimental setup.** We follow Xu et al. [42] and use a 70%-15%-15% (train-val-test) temporal split for all datasets. We adopt average precision (AP) as the performance metric. We also analyze separately predictions involving only nodes seen during training (transductive), and those involving

Table 1: **Average Precision** (AP) results for link prediction. We denote the best-performing model (highest mean AP) in **blue**. In 5 out of 6 datasets, PINT achieves the highest AP in the transductive setting. For the inductive case, PINT outperforms previous MP-TGNs and competes with CAW. We also evaluate PINT w/ and w/o relative positional features. Adopting positional features leads to significant performance gains.

| | Model | Reddit | Wikipedia | Twitter | UCI | Enron | LastFM |
|---|---|---|---|---|---|---|---|
| **Transductive** | GAT | $97.33 \pm 0.2$ | $94.73 \pm 0.2$ | - | - | - | - |
| | GraphSAGE | $97.65 \pm 0.2$ | $93.56 \pm 0.3$ | - | - | - | - |
| | Jodie | $97.11 \pm 0.3$ | $94.62 \pm 0.5$ | $98.23 \pm 0.1$ | $86.73 \pm 1.0$ | $77.31 \pm 4.2$ | $69.32 \pm 1.0$ |
| | DyRep | $97.98 \pm 0.1$ | $94.59 \pm 0.2$ | $98.48 \pm 0.1$ | $54.60 \pm 3.1$ | $77.68 \pm 1.6$ | $69.24 \pm 1.4$ |
| | TGAT | $98.12 \pm 0.2$ | $95.34 \pm 0.1$ | $98.70 \pm 0.1$ | $77.51 \pm 0.7$ | $68.02 \pm 0.1$ | $54.77 \pm 0.4$ |
| | TGN-Att | $98.70 \pm 0.1$ | $98.46 \pm 0.1$ | $98.00 \pm 0.1$ | $80.40 \pm 1.4$ | $79.91 \pm 1.3$ | $80.69 \pm 0.2$ |
| | CAW | $98.39 \pm 0.1$ | $98.63 \pm 0.1$ | $98.72 \pm 0.1$ | $92.16 \pm 0.1$ | $\mathbf{92.09 \pm 0.7}$ | $81.29 \pm 0.1$ |
| | **PINT** (w/o pos. feat.) | $98.62 \pm .04$ | $98.43 \pm .04$ | $98.53 \pm 0.1$ | $92.68 \pm 0.5$ | $83.06 \pm 2.1$ | $81.35 \pm 1.6$ |
| | **PINT** | $\mathbf{99.03 \pm .01}$ | $\mathbf{98.78 \pm 0.1}$ | $\mathbf{99.35 \pm .01}$ | $\mathbf{96.01 \pm 0.1}$ | $88.71 \pm 1.3$ | $\mathbf{88.06 \pm 0.7}$ |
| **Inductive** | GAT | $95.37 \pm 0.3$ | $91.27 \pm 0.4$ | - | - | - | - |
| | GraphSAGE | $96.27 \pm 0.2$ | $91.09 \pm 0.3$ | - | - | - | - |
| | Jodie | $94.36 \pm 1.1$ | $93.11 \pm 0.4$ | $96.06 \pm 0.1$ | $75.26 \pm 1.7$ | $76.48 \pm 3.5$ | $80.32 \pm 1.4$ |
| | DyRep | $95.68 \pm 0.2$ | $92.05 \pm 0.3$ | $96.33 \pm 0.2$ | $50.96 \pm 1.9$ | $66.97 \pm 3.8$ | $82.03 \pm 0.6$ |
| | TGAT | $96.62 \pm 0.3$ | $93.99 \pm 0.3$ | $96.33 \pm 0.1$ | $70.54 \pm 0.5$ | $63.70 \pm 0.2$ | $56.76 \pm 0.9$ |
| | TGN-Att | $97.55 \pm 0.1$ | $97.81 \pm 0.1$ | $95.76 \pm 0.1$ | $74.70 \pm 0.9$ | $78.96 \pm 0.5$ | $84.66 \pm 0.1$ |
| | CAW | $97.81 \pm 0.1$ | $\mathbf{98.52 \pm 0.1}$ | $\mathbf{98.54 \pm 0.4}$ | $92.56 \pm 0.1$ | $\mathbf{91.74 \pm 1.7}$ | $85.67 \pm 0.5$ |
| | **PINT** (w/o pos. feat.) | $97.22 \pm 0.2$ | $97.81 \pm 0.1$ | $96.10 \pm 0.1$ | $90.25 \pm 0.3$ | $75.99 \pm 2.3$ | $88.44 \pm 1.1$ |
| | **PINT** | $\mathbf{98.25 \pm .04}$ | $98.38 \pm .04$ | $98.20 \pm .03$ | $\mathbf{93.97 \pm 0.1}$ | $81.05 \pm 2.4$ | $\mathbf{91.76 \pm 0.7}$ |

novel nodes (inductive). We report mean and standard deviation of the AP over ten runs. For further details, see Appendix D. We provide additional results in the supplementary material.

**Results.** Table 1 shows that PINT is the best-performing method on five out of six datasets for the transductive setting. Notably, the performance gap between PINT and TGN-Att amounts to over 15% AP on UCI. The gap is also relatively high compared to CAW on LastFM, Enron, and UCI; with CAW being the best model only on Enron. We also observe that many models achieve relatively high AP on the attributed networks (Reddit, Wikipedia, and Twitter). This aligns well with findings from [38], where TGN-Att was shown to have competitive performance against CAW on Wikipedia and Reddit. The performance of GAT and TGAT (static GNNs) on Reddit and Wikipedia reinforces the hypothesis that the edge features add significantly to the discriminative power. On the other hand, PINT and CAW, which leverage relative identities, show superior performance relative to other methods when only time and degree information is available, i.e., on unattributed networks (UCI, Enron, and LastFM). Table 1 also shows the effect of using relative positional features. While including these features boosts PINT's performance systematically, our ablation study shows that PINT w/o positional features still outperforms other MP-TGNs on unattributed networks. In the inductive case, we observe a similar behavior: PINT is consistently the best MP-TGN, and is better than CAW on 3/6 datasets. Overall, PINT (w/ positional features) also yields the lowest standard deviations. This suggests that positional encodings might be a useful inductive bias for TGNs.

**Time comparison.** Figure 7 compares the training times of PINT against other TGNs. For fairness, we use the same architecture (number of layers & neighbors) for all MP-TGNs: i.e., the best-performing PINT. For CAW, we use the one that yielded results in Table 1. As expected, TGAT is the fastest model. Note that the average time/epoch of PINT gets amortized since positional features are pre-computed. Without these features, PINT's runtime closely matches TGN-Att. When trained for over 25 epochs, PINT runs considerably faster than CAW. We provide additional details and results in the supplementary material.

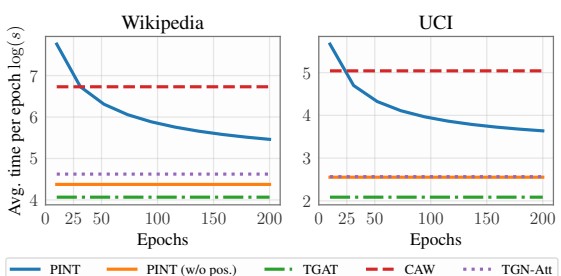

Figure 7: Time comparison: PINT versus TGNs (in log-scale). The cost of pre-computing positional features is quickly diluted as the number of epochs increases.

Table 2: Average precision results for TGN-Att + relative positional features.

| | Transductive | | | Inductive | | |
| --- | --- | --- | --- | --- | --- | --- |
| | UCI | Enron | LastFM | UCI | Enron | LastFM |
| TGN-Att | $80.40 \pm 1.4$ | $79.91 \pm 1.3$ | $80.69 \pm 0.2$ | $74.70 \pm 0.9$ | $78.96 \pm 0.5$ | $84.66 \pm 0.1$ |
| TGN-Att + RPF | $95.64 \pm 0.1$ | $85.04 \pm 2.5$ | $89.41 \pm 0.9$ | $92.82 \pm 0.4$ | $76.27 \pm 3.4$ | $91.63 \pm 0.3$ |
| PINT | $96.01 \pm 0.1$ | $88.71 \pm 1.3$ | $88.06 \pm 0.7$ | $93.97 \pm 0.1$ | $81.05 \pm 2.4$ | $91.76 \pm 0.7$ |

**Incorporating relative positional features into MP-TGNs.** We can use our relative positional features (RPF) to boost MP-TGNs. Table 2 shows the performance of TGN-Att with relative positional features on UCI, Enron, and LastFM. Notably, TGN-Att receives a significant boost from our RPF. However, PINT still beats TGN-Att+RPF on 5 out of 6 cases. The values for TGN-Att+RPF reflect outcomes from 5 repetitions. We have used the same model selection procedure as TGN-Att in Table 1, and incorporated $d = 4$-dimensional positional features

**Dimensionality of relative positional features.** We assess the performance of PINT as a function of the dimension $d$ of the relative positional features. Figure 8 shows the performance of PINT for $d \in \{4, 10, 15, 20\}$ on UCI and Enron. We report mean and standard deviation of AP on test set obtained from five independent runs. In all experiments, we re-use the optimal hyperparameters found with $d = 4$. Increasing the dimensionality of the positional features leads to performance gains on both datasets. Notably,

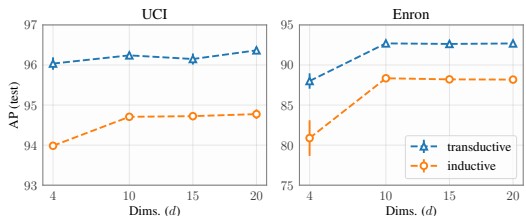

Figure 8: PINT: AP (mean and std) as a function of the dimensionality of the positional features.

we obtain a significant boost for Enron with $d = 10$: $92.69 \pm 0.09$ AP in the transductive setting and $88.34 \pm 0.29$ in the inductive case. Thus, PINT becomes the best-performing model on Enron (transductive). On UCI, for $d = 20$, we obtain $96.36 \pm 0.07$ and $94.77 \pm 0.12$ (inductive).

# 6 Conclusion

We laid a rigorous theoretical foundation for TGNs, including the role of memory modules, relationship between classes of TGNs, and failure cases for MP-TGNs. Together, our theoretical results shed light on the representational capabilities of TGNs, and connections with their static counterparts. We also introduced a novel TGN method, provably more expressive than the existing TGNs.

Key practical takeaways from this work: (a) temporal models should be designed to have injective update rules and to exploit both neighborhood and walk aggregation, and (b) deep architectures can likely be made more compute-friendly as the role of memory gets diminished with depth, provably.

## Acknowledgments and Disclosure of Funding

This work was supported by the Academy of Finland (Flagship programme: Finnish Center for Artificial Intelligence FCAI and 341763), ELISE Network of Excellence Centres (EU Horizon:2020 grant agreement 951847) and UKRI Turing AI World-Leading Researcher Fellowship, EP/W002973/1. We also acknowledge the computational resources provided by the Aalto Science-IT Project from Computer Science IT. AS and DM also would like to thank Jorge Perez, Jou-Hui Ho, and Hojin Kang for valuable discussions about TGNs, and the latter's input on a preliminary version of this work.

## Societal and broader impact

Temporal graph networks have shown remarkable performance in relevant domains such as social networks, e-commerce, and drug discovery. In this paper, we establish fundamental results that delineate the representational power of TGNs. We expect that our findings will help declutter the literature and serve as a seed for future developments. Moreover, our analysis culminates with PINT, a method that is provably more powerful than the prior art and shows superior predictive performance on several benchmarks. We believe that PINT (and its underlying concepts) will help engineers and researchers build better recommendation engines, improving the quality of systems that permeate our lives. Also, we do not foresee any negative societal impact stemming directly from this work.

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
