# Provably expressive temporal graph networks (Supplementary material)

## A  Further details on temporal graph networks

In this section we present more details about the models TGAT, TGN-Att, and CAW.

### A.1  Temporal graph attention (TGAT)

Temporal graph attention networks [42] combine time encoders [15] and self-attention [33]. In particular, the time encoder $\phi$ is given by

$$\phi(t - t') = [\cos(\omega_1(t - t') + b_1), \dots, \cos(\omega_d(t - t') + b_d)], \tag{S1}$$

where $\omega_i$'s and $b_i$'s are learned scalar parameters. The time embeddings are concatenated to the edge features before being fed into a typical self-attention layer, where the query $q$ is a function of a reference node $v$, and both values $V$ and keys $K$ depend on $v$'s temporal neighbors. Formally, TGAT first computes a matrix $C_v^{(\ell)}(t)$ whose $u$-th row is $c_{vu}^{(\ell)}(t) = [h_u^{(\ell-1)}(t) \parallel \phi(t - t_{uv}) \parallel e_{uv}]$ for all $(u, e_{uv}, t_{uv}) \in \mathcal{N}(v, t)$. Then, the output $\tilde{h}_v^{(\ell)}(t)$ of the AGG$^{(\ell)}$ function is given by

$$q = [h_v^{(\ell-1)}(t) \parallel \phi(0)]W_q^{(\ell)} \quad K = C_v^{(\ell)}(t)W_K^{(\ell)} \quad V = C_v^{(\ell)}(t)W_V^{(\ell)} \tag{S2}$$

$$\tilde{h}_v^{(\ell)}(t) = \mathrm{softmax}\left(qK^\top\right)V \tag{S3}$$

where $W_q^{(\ell)}, W_K^{(\ell)}$, and $W_V^{(\ell)}$ are model parameters. Regarding the UPDATE function, TGAT applies a multilayer perceptron, i.e., $h_v^{(\ell)}(t) = \mathrm{MLP}^{(\ell)}(h_v^{(\ell-1)}(t) \parallel \tilde{h}_v^{(\ell)}(t))$.

### A.2  Temporal graph networks with attention (TGN-Att)

We now discuss details regarding the MP-TGN framework omitted from the main paper for simplicity.

For the sake of generality, Rossi et al. [27] present a formulation for MP-TGNs that can handle node-level events, e.g., node feature updates. These events lead to $i$) updating node memory states, and $ii$) using the time-evolving node features as additional inputs for the message-passing functions. Nonetheless, to the best of our knowledge, all relevant CTDG benchmarks comprise only edge events. Therefore, for ease of presentation, we omit node events and temporal node features from our treatment. In Appendix E, we discuss how to handle node-level events.

Note that MP-TGNs update memory states only after an event occurs, otherwise it would incur information leakage. Unless we use the updated states to predict another event later on in the batch, this means that there might be no signal to propagate through memory modules. To get around this problem, Rossi et al. [27] propose updating the memory with messages coming from previous batches, and then predicting the interactions.

To speed up computations, MP-TGNs employ a form of batch learning where events in a same batch are aggregated. In our analysis, we assume that two events belong to the same batch only if they occur at the same timestamp. Importantly, memory aggregators allow removing ambiguity in the way the memory of a node participating in multiple events (at the same timestamp) is updated — without memory aggregator, two events involving a given node $i$ at the same time could lead to different ways of updating the state of $i$.

Suppose the event $\gamma = (i, u, t)$ occurs. MP-TGNs proceed by computing a memory-message function $\text{MEMMSG}_e$ for each endpoint of $\gamma$, i.e.,

$$m_{i,u}(t) = \text{MEMMSG}_e(s_i(t), s_u(t), t - t_i, e_{iu}(t))$$
$$m_{u,i}(t) = \text{MEMMSG}_e(s_u(t), s_i(t), t - t_u, e_{iu}(t))$$

Following the original formulation, we assume an identity memory-message function — simply the concatenation of the inputs, i.e., $\text{MEMMSG}_e(s_i(t), s_u(t), t - t_i, e_{iu}(t)) = [s_i(t), s_u(t), t - t_i, e_{iu}(t)]$.

Now, suppose two events $(i, u, t)$ and $(i, v, t)$ happen. MP-TGNs aggregate the memory-messages from these events using a function $\text{MEMAGG}$ to obtain a single memory message for $i$:

$$m_i(t) = \text{MEMAGG}(m_{i,u}(t), m_{i,v}(t))$$

Rossi et al. [27] propose non-learnable memory aggregators, such as the mean aggregator (average all memory messages for a given node), that we denote as $\text{MEANAGG}$ and adopt throughout our analysis. As an example, under events $(i, u, t)$ and $(i, v, t)$, the aggregated message for $i$ is $m_i(t) = 0.5([s_i(t), s_u(t), t - t_i, e_{iu}(t)] + [s_i(t), s_v(t), t - t_i, e_{iv}(t)])$.

The memory update of our query node $i$ is given by

$$s_i(t^+) = \text{MEMUPDATE}(s_i(t), m_i(t)).$$

Finally, we note that TGAT does not have a memory module. TGN-Att consists of the model resulting from augmenting TGAT with a GRU-based memory.

### A.3 Causal anonymous walks (CAW)

We now provide details regarding how CAW obtains edge embeddings for a query event $\gamma = (u, v, t)$.

A temporal walk is represented as $W = ((w_1, t_1), (w_2, t_2), \ldots, (w_L, t_L))$, with $t_1 > t_2 > \cdots > t_L$ and $(w_{i-1}, w_i, t_i) \in \mathcal{G}(t)$ for all $i > 1$. We denote by $S_u(t)$ the set of maximal temporal walks starting at $u$ of size at most $L$ obtained from the temporal graph at time $t$. Following the original paper, we drop the time dependence henceforth.

A given walk $W$ gets anonymized through replacing each element $w_i$ belonging to $W$ by a 2-element set of vectors $I_{\text{CAW}}(w_i; S_u, S_v)$ accounting for how many times $w_i$ appears at each position of walks in $S_u$ and $S_v$. These vectors are denoted by $g(w_i, S_u)$ and $g(w_i, S_v)$. The walk is encoded using a RNN:

$$\text{ENC}(W; S_u, S_v) = \text{RNN}([f_1(I_{\text{CAW}}(w_i; S_u, S_v))\|f_2(t_i - t_{i-1})]_{i=1}^L),$$

where $t_1 = t_0 = t$ and $f_1$ is

$$f_1(I_{\text{CAW}}(w_i; S_u, S_v)) = \text{MLP}(g(w_i, S_u)) + \text{MLP}(g(w_i, S_v)).$$

We note that the MLPs share parameters. The function $f_2$ is given by

$$f_2(t) = [\cos(\omega_i t), \sin(\omega_1 t), \ldots, \cos(\omega_d t), \sin(\omega_d t)]$$

where $\omega_i$'s are learned parameters.

To compute the embedding $h_\gamma$ for $(u, v, t)$, CAW considers two readout functions: mean and self-attention. Finally, the final link prediction is obtained from a 2-layer MLP over $h_\gamma$.

# B Proofs

## B.1 Further definitions and Lemmata

**Definition B1** (Monotone walk.)**.** *An $N$-length monotone walk in a temporal graph $\mathcal{G}(t)$ is a sequence $(w_1, t_1, w_2, t_2, \ldots, w_{N+1})$ such that $t_i > t_{i+1}$ and $(w_i, w_{i+1}, t_i) \in \mathcal{G}(t)$ for all $i$.*

**Definition B2** (Temporal diameter.)**.** *We say the temporal diameter of a graph $\mathcal{G}(t)$ is $\Delta$ if the longest monotone walk in $\mathcal{G}(t)$ has length (i.e, number of edges) exactly $\Delta$.*

**Lemma B1.** *If the TCTs of two nodes are isomorphic, then their monotone TCTs ([Definition 2](#)) are also isomorphic, i.e., $T_u(t) \cong T_v(t) \Rightarrow \tilde{T}_u(t) \cong \tilde{T}_v(t)$ for two nodes $u$ and $v$ of a dynamic graph.*

*Proof.* Since $T_u(t) \cong T_v(t)$, we have that

$$p = (u_0, t_1, u_1, t_2, u_2, \ldots) \text{ from } T_u(t) \iff p' = (f(u_0), t_1, f(u_1), t_2, f(u_2), \ldots) \text{ from } T_v(t),$$
$$\text{with } s_{u_i} = s_{f(u_i)} \text{ and } e_{u_i u_{i+1}}(t_{i+1}) = e_{f(u_i)f(u_{i+1})}(t_{i+1}) \text{ and } k_{u_i} = k_{f(u_i)} \quad \forall i$$

where $f : V(T_u(t)) \to V(T_v(t))$ is a bijection.

Assume that $\tilde{T}_u(t) \ncong \tilde{T}_v(t)$. Then, either there exists a path $p_s = (u_0', t_1', u_1', t_2', \ldots)$ in $\tilde{T}_u(t)$, such that $t_{k+1}' < t_k'$ for all $k$ (i.e., a monotone walk), with no corresponding one in $\tilde{T}_v(t)$ or vice-versa. Without loss of generality, let us consider the former case.

We can construct the path $p_s'$ in $T_v(t)$ by applying $f$ in all elements of $p_s$, i.e., $p_s' = (f(u_0'), t_1', f(u_1'), t_2', \ldots)$. Note that $p_s'$ is a monotone walk in $T_v(t)$. Since $\tilde{T}_v(t)$ is the maximal monotone subtree of $T_v(t)$, it must contain $p_s'$, leading to contradiction. $\qquad\square$

**Lemma B2.** *Let $\mathcal{G}(t)$ and $\mathcal{G}'(t)$ be any two non-isomorphic temporal graphs. If an MP-TGN obtains different multisets of node embeddings for $\mathcal{G}(t)$ and $\mathcal{G}'(t)$. Then, the temporal WL test decides $\mathcal{G}(t)$ and $\mathcal{G}'(t)$ are not isomorphic.*

*Proof.* Recall [Proposition 3](#) shows that if an MP-TGN with memory is able to distinguish two nodes, then there is a memoryless MP-TGN with $\Delta$ (temporal diameter) additional layers that does the same. Thus, it suffices to show that if the multisets of colors from temporal WL for $\mathcal{G}(t)$ and $\mathcal{G}'(t)$ after $\ell$ iterations are identical, then the multisets of embeddings from the memoryless MP-TGN are also identical, i.e., if $\{\!\{c^\ell(u)\}\!\}_{u \in V(\mathcal{G}(t))} = \{\!\{c^\ell(u')\}\!\}_{u' \in V(\mathcal{G}'(t))}$, then $\{\!\{h_u^{(\ell)}(t)\}\!\}_{u \in V(\mathcal{G}(t))} = \{\!\{h_{u'}^{(\ell)}(t)\}\!\}_{u' \in V(\mathcal{G}'(t))}$. To do so, we repurpose the proof of Lemma 2 in [43].

More broadly, we show that for any two nodes of a temporal graph $\mathcal{G}(t)$, if the temporal WL returns $c^\ell(u) = c^\ell(v)$, we have that corresponding embeddings from MP-TGN without memory are identical $h_u^\ell(t) = h_v^\ell(t)$. We proceed with a proof by induction.

[*Base case*] For $\ell = 0$, the proposition trivially holds as the temporal WL has the initial node features as colors, and memoryless MP-TGNs have these features as embeddings.

[*Induction step*] Assume the proposition holds for iteration $\ell$. Thus, for any two nodes $u, v$, if $c^{\ell+1}(u) = c^{\ell+1}(v)$, we have

$$(c^\ell(u), \{\!\{(c^\ell(i), e_{iu}(t'), t') : (u, i, t') \in \mathcal{G}(t)\}\!\}) = (c^\ell(v), \{\!\{(c^\ell(j), e_{jv}(t'), t') : (v, j, t') \in \mathcal{G}(t)\}\!\})$$

and, by the induction hypothesis, we know

$$(h_u^{(\ell)}(t), \{\!\{(h_i^{(\ell)}(t), e_{iu}(t'), t') : (u, i, t') \in \mathcal{G}(t)\}\!\}) =$$
$$(h_v^{(\ell)}(t), \{\!\{(h_j^{(\ell)}(t), e_{jv}(t'), t') : (v, j, t') \in \mathcal{G}(t)\}\!\})$$

We also note that this last identity also implies

$$(h_u^{(\ell)}(t), \{\!\{(h_i^{(\ell)}(t), t - t', e) \mid (i, e, t') \in \mathcal{N}(u, t)\}\!\}) =$$
$$(h_v^{(\ell)}(t), \{\!\{(h_j^{(\ell)}(t), t - t', e) \mid (j, e, t') \in \mathcal{N}(v, t)\}\!\})$$

since there exists an event $(u, i, t') \in \mathcal{G}(t)$ with feature $e_{ui}(t') = e$ iff there is an element $(i, e, t') \in \mathcal{N}(u, t)$.

As a result, the inputs of the MP-TGN's aggregation and update functions are identical, which leads to identical outputs $h_u^{(\ell+1)}(t) = h_v^{(\ell+1)}(t)$. Therefore, if the temporal WL test obtains identical multisets of colors for two temporal graphs after $\ell$ steps, the multisets of embeddings at layer $\ell$ for these graphs are also identical. $\qquad\square$

**Lemma B3** (Lemma 5 in [43]). *Assume $\mathcal{X}$ is countable. There exists a function $f : \mathcal{X} \to \mathbb{R}^n$ so that $h(X) = \sum_{x \in X} f(x)$ is unique for each multiset $X \subset \mathcal{X}$ of bounded size. Moreover, any multiset function $g$ can be decomposed as $g(X) = \varphi\left(\sum_{x \in X} f(x)\right)$ for some function $\varphi$.*

### B.2 Proof of Proposition 1: Relationship between DTDGs and CTDGs

*Proof.* We prove the two statements in Proposition 1 separately. In the following, we treat CTDGs as sets of events up to a given timestamp.

**Statement 1:** For any DTDG we can build a CTDG that contains the same information.

A DTDG consists of a sequence of graphs with no temporal information. We can model this using the CTDG formalism by setting a fixed time difference $\delta$ between consecutive elements $\mathsf{G}(t_i), \mathsf{G}(t_{i+1})$ of the CTDG, i.e., $t_{i+1} - t_i = \delta$ for all $i \geq 0$.

Consider a DTDG given by the sequence $(G_1, G_2, \dots)$. To build the equivalent CTDG, we define $S(G_i)$ as the set of edge events corresponding to $G_i$, i.e., $S(G_i) = \{(u, v, i\delta) : (u, v) \in E(G_i)\}$. We also make the edge features of these events match those in the DTDG, i.e., $e_{uv}(i\delta) = e_{uv} \in \mathcal{E}_i$. To account for node features, for all $u \in V(G_i)$, we create an event $(u, \diamond, i\delta)$ between $u$ and a dummy node $\diamond$, with feature $e_{u\diamond}(i\delta) = x_u \in \mathcal{X}_i$. Let $C(G_i)$ denote the set comprising these node-level events. Then, we can construct the CTDG $\mathsf{G}(t_i) = \cup_{j=1}^i S(G_j) \cup C(G_j)$ for $i = 1, \dots$. Reconstructing the DTDG $(G_1, G_2, \dots)$ is trivial. To build $G_i$, it suffices to select all events at time $i\delta$ in the CTDG. Events involving $\diamond$ determine node features and the remaining ones constitute edges in the DTDG.

**Statement 2:** The converse holds if the CTDG timestamps form a subset of some uniformly spaced countable set.

We say that a countable set $A \subset \mathbb{R}$ is uniformly spaced if there exists some $\delta \in \mathbb{R}$ such that $a_{i+1} - a_i = \delta$ for all $i$ where $(a_1, a_2, \dots)$ is the ordered sequence formed from elements $a_r$ of $A$, i.e., $a_1 < a_2 < \dots < a_i < a_{i+1}, \dots$

Note that DTDGs are naturally represented by a set of uniformly spaced timestamps. This is because DTDGs correspond to sequences that do not contain any time information. Let us denote the set of CTDG timestamps $T \subseteq \mathcal{T}$ such that $\mathcal{T}$ is countable and uniformly spaced. Our idea is to construct a DTDG sequence with timestamps that coincide with the elements in $\mathcal{T}$. Then, since $T \subseteq \mathcal{T}$, we do not lose any information pertaining to events occurring at timestamps given by $T$. Without loss of generality, in the following we assume that the elements of $T$ and $\mathcal{T}$ are arranged in their increasing order respectively, i.e., $t_i < t_{i+1}$ for all $i$, and $\tau_k < \tau_{k+1}$ for all $k$.

Consider a CTDG $(\mathsf{G}(t_1), \mathsf{G}(t_2), \dots)$ such that $\mathsf{G}(t_i) = \{(u, v, t) : t \in T \text{ and } t \leq t_i\}$ for $t_i \in T$. Also, let us denote $H(t_i) = \{(u, v, t) \in \mathsf{G}(t_i) : t = t_i\}$ the set of events at time $t_i \in T$. We can build a corresponding DTDG $(G_1, G_2, \dots)$ such that for all $\tau_k \in \mathcal{T}$ the $k$-th snapshot $G_k$ is

$$V(G_k) = \begin{cases} \{u : (u, \cdot, \tau_k) \in H(\tau_k)\}, & \text{if } \tau_k \in T; \\ \emptyset, & \text{otherwise.} \end{cases}$$

$$E(G_k) = \begin{cases} \{(u, v) : (u, v, \tau_k) \in H(\tau_k)\}, & \text{if } \tau_k \in T; \\ \emptyset, & \text{otherwise.} \end{cases}$$

To recover the original CTDG, we can adapt the reconstruction procedure we used in the previous part of the proof. We define

$$\tilde{I} = \{(i, k) \in \mathbb{N} \times \mathbb{N} : \tau_k = t_i \text{ for } t_i \in T \text{ and } \tau_k \in \mathcal{T}\}. \tag{S4}$$

Note that we can treat $\tilde{I}$ as a map by defining $\tilde{I}(i) = k$ if and only if $(i, k) \in \tilde{I}$. To recover the original CTDG, we first create the set of events $S(G_k) = \{(u, v, k\delta) : (u, v) \in E(G_k)\}$. Then, we build $\mathsf{G}(t_i) = \cup_{j:j \leq \tilde{I}(i)} S(G_j)$ for $t_i \in T$. $\qquad\square$

### B.3 Proof of Lemma 1

*Proof.* Here we show that if two nodes $u$ and $v$ have isomorphic ($L$-depth) TCTs, then MP-TGNs (with $L$-layers) compute identical embeddings for $u$ and $v$. Formally, let $T_{u,\ell}(t)$ denote the TCT of $u$ with $\ell$ layers. We want to show that $T_{u,\ell}(t) \cong T_{v,\ell}(t) \Rightarrow h_u^{(\ell)}(t) = h_v^{(\ell)}(t)$. We employ a proof by induction on $\ell$. Since there is no ambiguity, we drop the dependence on time in the following.

[*Base case*] Consider $\ell = 1$. By the isomorphism assumption $T_{u,1} \cong T_{v,1}$, $h_u^{(0)} = s_u = s_v = h_v^{(0)}$ — roots of both trees have the same states. Also, for any children $i$ of $u$ in $T_{u,1}$ there is a corresponding one $f(i)$ in $T_{v,1}$ with $s_i = s_{f(i)}$. Recall that the $\ell$-th layer aggregation function $\text{AGG}^{(\ell)}(\cdot)$ acts on multisets of triplets of previous-layer embeddings, edge features and timestamps of temporal neighbors (see Equation 1). Since the temporal neighbors of $u$ correspond to its children in $T_{u,1}$, then the output of the aggregation function for $u$ and $v$ are identical: $\tilde{h}_u^{(1)} = \tilde{h}_v^{(1)}$. In addition, since the initial embeddings of $u$ and $v$ are also equal (i.e., $h_u^{(0)} = h_v^{(0)}$), we can ensure that the update function returns $h_u^{(1)} = h_v^{(1)}$.

[*Induction step*] Assuming that $T_{u,\ell-1} \cong T_{v,\ell-1} \Rightarrow h_u^{(\ell-1)} = h_v^{(\ell-1)}$ for any pair of nodes $u$ and $v$, we will show that $T_{u,\ell} \cong T_{v,\ell} \Rightarrow h_u^{(\ell)} = h_v^{(\ell)}$. For any children $i$ of $u$, let us define the subtree of $T_{u,\ell}$ rooted at $i$ by $T_i$. We know that $T_i$ has depth $\ell - 1$, and since $T_{u,\ell} \cong T_{v,\ell}$, there exists a corresponding subtree of $T_v$ (of depth $\ell - 1$) rooted at $f(i)$ such that $T_i \cong T_{f(i)}$. Using the induction hypothesis, we obtain that the multisets of embeddings from the children $i$ of $u$ and children $f(i)$ of $v$ are identical. Note that if two $\ell$-depth TCTs are isomorphic, they are also isomorphic up to depth $\ell - 1$, i.e., $T_{u,\ell} \cong T_{v,\ell}$ implies $T_{u,\ell-1} \cong T_{v,\ell-1}$ and, consequently, $h_u^{(\ell-1)} = h_v^{(\ell-1)}$ (by induction hypothesis). Thus, the input of the aggregation and update functions are identical and they compute the same embeddings for $u$ and $v$. $\square$

### B.4 Proof of Proposition 2: MP-TGNs with injective message passing

*Proof.* Consider MP-TGNs with parameter values that make $\text{AGG}^{(\ell)}(\cdot)$ and $\text{UPDATE}^{(\ell)}(\cdot)$ injective functions on multisets of triples of hidden representations, edge features and timestamps. The existence of these parameters is guaranteed by the fact that, at any given time $t$, the space of node states (and hidden embeddings from temporal neighbors), edge features and timestamps is finite (see Lemma B3).

Again, let $T_{u,\ell}(t)$ denote the TCT of $u$ with $\ell$ layers. We want to prove that, under the injectivity assumption, if $T_{u,\ell}(t) \not\cong T_{v,\ell}(t)$, then $h_u^{(\ell)}(t) \neq h_u^{(\ell)}(t)$ for any two nodes $u$ and $v$. In the following, we simplify notation by removing the dependence on time. We proceed with proof by induction on the TCT's depth $\ell$. Also, keep in mind that $\varphi_\ell = \text{UPDATE}^{(\ell)} \circ \text{AGG}^{(\ell)}$ is injective for any $\ell$.

[*Base case*] For $\ell = 1$, if $T_{u,1} \not\cong T_{v,1}$ then the root node states are different (i.e., $s_u \neq s_v$) or the multiset of states/edge features/ timestamps triples from $u$ and $v$'s children are different. In both cases, the inputs of $\varphi_\ell$ are different and it therefore outputs different embeddings for $u$ and $v$.

[*Induction step*] The inductive hypothesis is $T_{u,\ell-1} \not\cong T_{v,\ell-1} \Rightarrow h_u^{(\ell-1)} \neq h_v^{(\ell-1)}$ for any pair of nodes $u$ and $v$. If $T_{u,\ell} \not\cong T_{v,\ell}$, at least one of the following holds: i) the states of $u$ and $v$ are different, ii) the multisets of edges (edge features/ timestamps) with endpoints in $u$ and $v$ are different, or iii) there is no pair-wise isomorphism between the TCTs rooted at $u$ and $v$'s children. In the first two cases, $\varphi_\ell$ trivially outputs different embeddings for $u$ and $v$. We are left with the case in which only the latter occurs. Using our inductive hypothesis, the lack of a (isomorphism ensuring) bijection between the TCTs rooted at $u$ and $v$'s children implies there is also no bijection between their multiset of embeddings. In turn, this guarantees that $\varphi$ will output different embeddings for $u$ and $v$. $\square$

### B.5 Proof of Proposition 3: The role of memory

*Proof.* We prove the two parts of the proposition separately. In the following proofs, we rely on the concept of monotone TCTs (see Definition 2).

**Statement 1:** If $L < \Delta$: $\mathcal{Q}_L^{[M]}$ is strictly stronger than $\mathcal{Q}_L$.

We know that the family of $L$-layer MP-TGNs with memory comprises the family of $L$-layer MP-TGNs without memory (we can assume identity memory). Therefore, $\mathcal{Q}_L^{[M]}$ is at least as powerful as $\mathcal{Q}_L$. To show that $\mathcal{Q}_L^{[M]}$ is strictly stronger (more powerful) than $\mathcal{Q}_L$, when $L < \Delta$, it suffices to create an example for which memory can help distinguish a pair of nodes. We provide a trivial example in Figure S1 for $L = 1$. Note that the 1-depth TCTs of $u$ and $v$ are isomorphic when no memory is used. However, when equipped with memory, the interaction $(b, c, t_1)$ affects the states of $v$ and $c$, making the 1-depth TCTs of $u$ and $v$ (at time $t > t_2$) no longer isomorphic.

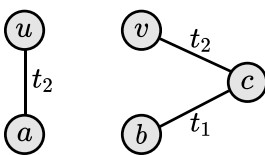

Figure S1: Temporal graph where all initial node features and edge features are identical, and $t_2 > t_1$.

**Statement 2:** For any $L : \mathcal{Q}_{L+\Delta}$ is at least as powerful as $\mathcal{Q}_L^{[M]}$.

It suffices to show that if $\mathcal{Q}_{L+\Delta}$ cannot distinguish a pair of nodes $u$ and $v$, $\mathcal{Q}_L^{[M]}$ cannot distinguish them too. Let $T_{u,L}^M(t)$ and $T_{u,L}(t)$ denote the $L$-depth TCTs of $u$ with and without memory respectively. Using Lemma 1, this is equivalent to showing that $T_{u,L+\Delta}(t) \cong T_{v,L+\Delta}(t) \Rightarrow T_{u,L}^M(t) \cong T_{v,L}^M(t)$, since no MP-TGN can separate nodes associated with isomorphic TCTs. In the following, when we omit the number of layers from TCTs, we assume TCTs of arbitrary depth.

**Step 1:** Characterizing the dependence of memory on initial states and events in the dynamic graph.

We now show that the memory for a node $u$, after processing all events with timestamp $\le t_n$, depends on the initial states of a set of nodes $\mathcal{V}_u^n$, and a set of events annotated with their respective timestamps and features $\mathcal{B}_u^n$. If at time $t_n$ no event involves a node $z$, we set $\mathcal{B}_z^n = \mathcal{B}_z^{n-1}$ and $\mathcal{V}_z^n = \mathcal{V}_z^{n-1}$. We also initialize $\mathcal{B}_u^0 = \emptyset$ and $\mathcal{V}_u^0 = \{u\}$ for all nodes $u$. We proceed with a proof by induction on the number of observed timestamps $n$.

[*Base case*] Let $\mathcal{I}_1(u) = \{v : (u, v, t_1) \in \mathcal{G}(t_1^+)\}$ be the set of nodes interacting with $u$ at time $t_1$, where $\mathcal{G}(t_1^+)$ is the temporal graph right after $t_1$. Similarly, let $\mathcal{J}_1(u) = \{(u, \cdot, t_1) \in \mathcal{G}(t_1^+)\}$ be the set of events involving $u$ at time $t_1$. Recall that up until $t_1$, all memory states equal initial node features (i.e., $s_u(t_1) = s_u(0)$). Then, the updated memory (see Equation 3 and Equation 4) for $u$ depends on $\mathcal{V}_u^1 = \mathcal{V}_u^0 \cup_{v \in \mathcal{I}(u)} \mathcal{V}_v^0, \mathcal{B}_u^1 = \mathcal{B}_u^0 \cup \mathcal{J}_1(u)$.

[*Induction step*] Assume that for timestamp $t_{n-1}$ the proposition holds. We now show that it holds for $t_n$. Since the proposition holds for $n-1$ timestamps, we know that the memory of any $w$ that interacts with $u$ in $t_n$, i.e. $w \in \mathcal{I}_n(u)$, depends on $\mathcal{V}_w^{n-1}$ and $\mathcal{B}_w^{n-1}$, and the memory of $u$ so far depends on $\mathcal{V}_u^{n-1}$ and $\mathcal{B}_u^{n-1}$. Then, the updated memory for $u$ depends on $\mathcal{V}_u^n = \mathcal{V}_u^{n-1} \cup_{w \in \mathcal{I}(u)} \{w, u\} \cup \mathcal{V}_w^{n-1}$ and $\mathcal{B}_u^n = \mathcal{B}_u^{n-1} \cup \mathcal{J}_n(u) \cup_{w \in \mathcal{I}(u)} \mathcal{B}_w^{n-1}$.

**Step 2:** $(z, w, t_{zw}) \in \mathcal{B}_u^n$ if and only if there is a path $(u_k, t_k = t_{zw}, u_{k+1})$ in $\tilde{T}_u(t_n^+)$ — the monotone TCT of $u$ (see Definition 2) after processing events with timestamp $\le t_n$ — with either $\sharp u_k = z, \sharp u_{k+1} = w$ or $\sharp u_k = w, \sharp u_{k+1} = z$.

[*Forward direction*] An event $(z, w, t_{zw})$ with $t_{zw} \le t_n$ will be in $\mathcal{B}_u^n$ only if $z = u$ or $w = u$, or if there is a subset of events $\{(u, \sharp u_1, t_1), (\sharp u_1, \sharp u_2, t_2), \dots, (\sharp u_k, \sharp u_{k+1}, t_{zw})\}$ with $\sharp u_k = z$ and $\sharp u_{k+1} = w$ such that $t_n \ge t_1 > \dots > t_{zw}$. In either case, this will lead to root-to-leaf path in $\tilde{T}_u(t_n^+)$ passing through $(u_k, t_{zw}, u_{k+1})$. This subset of events can be easily obtained by backtracking edges that caused unions/updates in the procedure from Step 1.

[*Backward direction*] Assume there is a subpath $p = (u_k, t_k = t_{zw}, u_{k+1}) \in \tilde{T}_u(t_n^+)$ with $\sharp u_k = z$ and $\sharp u_{k+1} = w$ such that $(z, w, t_{zw}) \notin \mathcal{B}_u^n$. Since we can obtain $p$ from $\tilde{T}_u(t_n^+)$, we know that the sequence of events $r = ((u, \sharp u_1, t_1), \dots, (\sharp u_{k-2}, \sharp u_{k-1} = z, t_{k-1}), (z, w, t_k = t_{zw}))$ happened and that $t_i > t_{i+1} \quad \forall i$. However, since $(z, w, t_{zw}) \notin \mathcal{B}_u^n$, there must be no monotone walk starting from $u$ going through the edge $(z, w, t_{zw})$ to arrive at $w$, which is exactly what $r$ characterizes. Thus, we reach contradiction.

Note that the nodes in $\mathcal{V}_u^n$ are simply the nodes that have an endpoint in the events $\mathcal{B}_u^n$, and therefore are also nodes in $\tilde{T}_u(t_n^+)$ and vice-versa.

**Step 3:** For any node $u$, there is a bijection that maps $(\mathcal{V}_u^n, \mathcal{B}_u^n)$ to $\tilde{T}_u(t_n^+)$.

First, we note that $(\mathcal{V}_u^n, \mathcal{B}_u^n)$ depends on a subset of all events, which we represent as $\mathcal{G}' \subseteq \mathcal{G}(t_n^+)$. Since $\mathcal{B}_u^n$ contains all events in $\mathcal{G}'$ and $(\mathcal{V}_u^n, \mathcal{B}_u^n)$ can be uniquely constructed from $\mathcal{G}'$, then there is a bijection $g$ that maps from $\mathcal{G}'$ to $(\mathcal{V}_u^n, \mathcal{B}_u^n)$.

Similarly, $\tilde{T}_u(t_n^+)$ also depends on a subset of events which we denote by $\mathcal{G}'' \subseteq \mathcal{G}(t_n^+)$. We note that the unique events in $\tilde{T}_u(t_n^+)$ correspond to $\mathcal{G}''$, and we can uniquely build the tree $\tilde{T}_u(t_n^+)$ from $\mathcal{G}''$. This implies that there is a bijection $h$ that maps from $\mathcal{G}''$ to $\tilde{T}_u(t_n^+)$.

Previously, we have shown that all events in $\mathcal{B}_u^n$ are also in $\tilde{T}_u(t_n^+)$ and vice-versa. This implies that both sets depend on the same events, and thus on the same subset of all events, i.e., $\mathcal{G}' = \mathcal{G}'' = \mathcal{G}_S$. Since there is a bijection $g$ between $\mathcal{G}_S$ and $(\mathcal{V}_u^n, \mathcal{B}_u^n)$, and a bijection $h$ between $\mathcal{G}_S$ and $\tilde{T}_u(t_n^+)$, there exists a bijection $f$ between $(\mathcal{V}_u^n, \mathcal{B}_u^n)$ and $\tilde{T}_u(t_n^+)$.

**Step 4:** If $T_{u,L+\Delta}(t^+) \cong T_{v,L+\Delta}(t^+)$, then $T_{u,L}^M(t^+) \cong T_{v,L}^M(t^+)$.

To simplify notation, we omit here the dependence on time.

Any node $w \in T_{u,L}^M$ also appears in $T_{u,L+\Delta}$ at the same level. The subtree of $T_{u,L+\Delta}$ rooted at $w$, denoted here by $T_w'$, has depth at least $k \geq \Delta$. Note that $T_w'$ corresponds to the $k$-depth TCT of $\sharp w$. Since the depth of $T_w'$ is at least $\Delta$, we know that the $\tilde{T}_w' \cong \tilde{T}_{\sharp w}$ — i.e., imposing time-constraints to $T_w'$ results in the monotone TCT of node $\sharp w$. Also, because the memory of $\sharp w$ depends on $\tilde{T}_{\sharp w}$, $T_w'$ comprises the information used to compute the memory state of $\sharp w$. Note that this applies to any $w$ in $T_{u,L}^M$; thus, $T_{u,L+\Delta}$ contains all we need to compute the states of any node of the dynamic graph that appears in $T_{u,L}^M$. The same argument applies to $T_{v,L+\Delta}$ and $T_{v,L}^M$. Finally, since $T_{u,L}^M$ can be uniquely computed from $T_{u,L+\Delta}$, and $T_{v,L}^M$ from $T_{v,L+\Delta}$, if $T_{u,L+\Delta} \cong T_{v,L+\Delta}$, then $T_{u,L}^M \cong T_{v,L}^M$. $\qquad\square$

## B.6 Proof of Proposition 4: Limitations of TGAT and TGN-Att

*Proof.* In this proof, we first provide an example of a dynamic graph where the TCTs of two nodes $u$ and $v$ are not isomorphic. Then, we show that we can not find a TGAT model such that $h_u^{(L)}(t) \neq h_v^{(L)}(t)$, i.e., TGAT does not distinguish $u$ and $v$. Next, we show that even if we consider TGATs with memory (TGN-Att), it is still not possible to distinguish nodes $u$ and $v$ in our example.

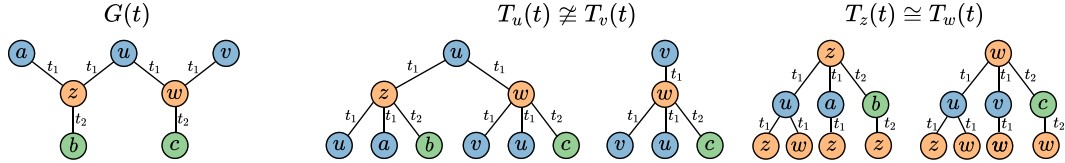

Figure S2: (**Leftmost**) Example of a temporal graph for which TGN-Att and TGAT cannot distinguish nodes $u$ and $v$ even though their TCTs are non-isomorphic. Colors denote node features and all edge features are identical, and $t_2 > t_1$ (and $t > t_2$). (**Right**) The 2-depth TCTs of nodes $u, v, z$ and $w$. The TCTs of $u$ and $v$ are non-isomorphic whereas the TCTs of $z$ and $w$ are isomorphic.

Figure S2(leftmost) provides a temporal graph where all edge events have the same edge features. Colors denote node features. As we can observe, the TCTs of nodes $u$ and $v$ are not isomorphic. In the following, we consider node distinguishability at time $t > t_2$.

**Statement 1:** TGAT cannot distinguish the nodes $u$ and $v$ in our example.

**Step 1:** For any TGAT with $\ell$ layers, we have that $h_w^{(\ell)}(t) = h_z^{(\ell)}(t)$.

We note that the $\ell$-layer TCTs of nodes $w$ and $z$ are isomorphic, for any $\ell$. To see this, one can consider the symmetry around node $u$ that allows us to define a node permutation function (bijection) $f$ given by $f(z) = w, f(w) = z, f(a) = v, f(u) = u, f(b) = c, f(c) = b, f(v) = a$. Figure S2(right) provides an illustration of the 2-depth TCTs of $z$ and $w$ at time $t > t_2$.

By Lemma 1, if the $\ell$-layer TCTs of two nodes $z$ and $w$ are isomorphic, then no $\ell$-layer MP-TGN can distinguish them. Thus, we conclude that $h_w^{(\ell)}(t) = h_z^{(\ell)}(t)$ for any TGAT with arbitrary number of layers $\ell$.

**Step 2:** There is no TGAT such that $h_v^{(\ell)}(t) \neq h_u^{(\ell)}(t)$.

To compute $h_v^{(\ell)}(t)$, TGAT aggregates the messages of $v$'s temporal neighbors at layer $\ell - 1$, and then combines $h_v^{(\ell-1)}(t)$ with the aggregated message $\tilde{h}_v^{(\ell-1)}(t)$ to obtain $h_v^{(\ell)}(t)$.

Note that $\mathcal{N}(u,t) = \{(z, e, t_1), (w, e, t_1)\}$ and $\mathcal{N}(v,t) = \{(w, e, t_1)\}$, where $e$ denotes an edge feature vector. Also, we have previously shown that $h_w^{(\ell-1)}(t) = h_z^{(\ell-1)}(t)$.

Using the TGAT aggregation layer (Equation S2), the query vectors of $u$ and $v$ are $q_u = [h_u^{(\ell-1)}(t)||\phi(0)]W_q^{(\ell)}$ and $q_v = [h_v^{(\ell-1)}(t)||\phi(0)]W_q^{(\ell)}$, respectively.

Since all events have the common edge features $e$, the matrices $C_u^{(\ell)}$ and $C_v^{(\ell)}$ share the same vector in their rows. The single-row matrix $C_v^{(\ell)}$ is given by $C_v^{(\ell)} = [h_w^{(\ell-1)}(t)||\phi(t-t_1)||e]$, while the two-row matrix $C_u^{(\ell)} = \left[ [h_w^{(\ell-1)}(t)||\phi(t-t_1)||e]; [h_z^{(\ell-1)}(t)||\phi(t-t_1)||e] \right]$, with $h_w^{(\ell-1)}(t) = h_z^{(\ell-1)}(t)$. We can express $C_u^{(\ell)} = [1,1]^\top r$ and $C_v^{(\ell)} = r$, where $r$ denotes the row vector $r = [h_z^{(\ell-1)}(t)||\phi(t-t_1)||e]$.

Using the key and value matrices of node $v$, i.e., $K_v = C_v^{(\ell)}W_K^{(\ell)}$ and $V_v = C_v^{(\ell)}W_V^{(\ell)}$, we have that

$$
\begin{aligned}
\tilde{h}_v^{(\ell)}(t) &= \text{softmax}(q_v K_v^\top)V_v \\
&= \underbrace{\text{softmax}(q_v K_v^\top)}_{=1} r W_V^{(\ell)} && \text{[softmax of a single element is 1]} \\
&= r W_V^{(\ell)} \\
&= \underbrace{\text{softmax}(q_u K_u^\top)[1,1]^\top}_{=1} r W_V^{(\ell)} = \tilde{h}_u^{(\ell)}(t) && \text{[softmax outputs a convex combination]}
\end{aligned}
$$

We have shown that the aggregated messages of nodes $u$ and $v$ are the same at any layer $\ell$. We note that the initial embeddings are also identical $h_v^{(0)}(t) = h_u^{(0)}(t)$ as $u$ and $v$ have the same color. Recall that the update step is $h_v^{(\ell)}(t) = \text{MLP}(h_v^{(\ell-1)}(t), \tilde{h}_v^{(\ell)}(t))$. Therefore, if the initial embeddings are identical, and the aggregated messages at each layer are also identical, we have that $h_u^{(\ell)}(t) = h_v^{(\ell)}(t)$ for any $\ell$.

**Statement 2:** TGN-Att cannot distinguish the nodes $u$ and $v$ in our example.

We now show that adding a memory module to TGAT produces node states such that $s_u(t) = s_v(t) = s_a(t)$, $s_z(t) = s_w(t)$, and $s_b(t) = s_c(t)$. If that is the case, then these node states could be treated as node features in a equivalent TGAT model of our example in Figure S2, proving that there is no TGN-Att such that $h_v^{(\ell)}(t) \neq h_u^{(\ell)}(t)$. In the following, we consider TGN-Att with average memory aggregators (see Appendix A).

We begin by showing that $s_a(t) = s_u(t) = s_v(t)$ after memory updates. We note that the memory message node $a$ receives is $[e||t_1||s_z(t_1)]$. The memory message node $u$ receives is $\text{MEANAGG}([e||t_1||s_w(t_1)], [e||t_1||s_z(t_1)])$, but since $s_w(t_1) = s_z(t_1)$, both messages are the same, and the average aggregator outputs $[e||t_1||s_z(t_1)]$. Finally, the message that node $v$ receives is $[e||t_1||s_w(t_1)] = [e||t_1||s_z(t_1)]$. Since all three nodes receive the same memory message and have the same initial features, their updated memory states are identical.

Now we show that $s_z(t) = s_w(t)$, for $t_1 < t \leq t_2$. Note that the message that node $z$ receives is $\text{MEANAGG}([e||t_1||s_a(t_1)], [e||t_1||s_u(t_1)]) = [e||t_1||s_u(t_1)]$, with $s_u(t_1) = s_a(t_1)$. The message that node $w$ receives is $\text{MEANAGG}([e||t_1||s_u(t_1)], [e||t_1||s_v(t_1)]) = [e||t_1||s_u(t_1)]$. Again, since the initial features and the messages received by each node are equal, $s_z(t) = s_w(t)$ for $t_1 < t \leq t_2$.

We can then use this to show that $s_z(t) = s_w(t)$ for $t > t_2$. Note that at time $t_2$, the message that nodes $z$ and $w$ receive are $[e||t_2 - t_1||s_b(t_2)]$ and $[e||t_2 - t_1||s_c(t_2)]$, respectively. Also, note that

$s_b(t_2) = s_c(t_2) = s_b(0) = s_c(0)$ as the states of $b$ and $c$ are only updated right after $t_2$. Because the received messages and the previous states (up until $t_2$) of $z$ and $w$ are identical, we have that $s_z(t) = s_w(t)$ for $t > t_2$.

Finally, we show that $s_b(t) = s_c(t)$. Using that $s_z(t_2) = s_w(t_2)$ in conjunction with the fact that node $b$ receives message $[[e\|t_2 - t_1\|s_z(t_2)]]$, and node $c$ receives $[e\|t_2 - t_1\|s_w(t_2)]$, we obtain $s_b(t) = s_c(t)$ since initial memory states and messages that the nodes received are the same. $\qquad\square$

## B.7 Proof of Proposition 5: Limitations of MP-TGN and CAWs

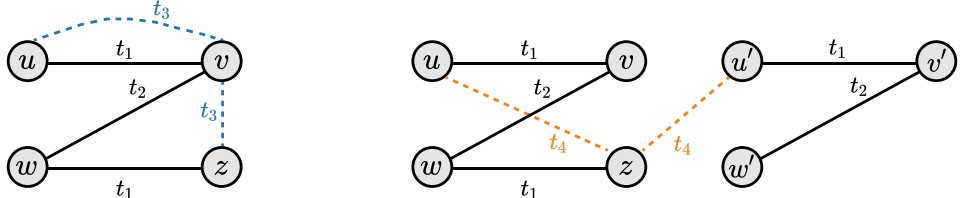

Figure S3: (**Left**) Example of a temporal graph for which CAW can distinguish the events $(u, v, t_3)$ and $(z, v, t_3)$ but MP-TGNs cannot. We assume that all edge and node features are identical, and $t_{k+1} > t_k$ for all $k$. (**Right**) Example for which MP-TGNs can distinguish $(u, z, t_4)$ and $(u', z, t_4)$ but CAW cannot.

*Proof.* Using the example in Figure S3(Left), we adapt a construction by Wang et al. [38] to show that CAW can separate events that MP-TGNs adopting node embedding concatenation cannot. We first note that the TCTs of $u$ and $z$ are isomorphic. Thus, since $v$ is a common endpoint in $(u, v, t_3)$ and $(z, v, t_3)$, no MP-TGN can distinguish these two events. Nonetheless, CAW obtains the following anonymized walks for the event $(u, v, t_3)$:

$$\underbrace{\{[1, 0, 0], [0, 1, 0]\}}_{I_{\text{CAW}}(u; S_u, S_v)} \xrightarrow{t_1} \underbrace{\{[0, 1, 0], [2, 0, 0]\}}_{I_{\text{CAW}}(v; S_u, S_v)}$$

$$\underbrace{\{[0, 1, 0], [2, 0, 0]\}}_{I_{\text{CAW}}(v; S_u, S_v)} \xrightarrow{t_1} \underbrace{\{[1, 0, 0], [0, 1, 0]\}}_{I_{\text{CAW}}(u; S_u, S_v)}$$

$$\underbrace{\{[0, 1, 0], [2, 0, 0]\}}_{I_{\text{CAW}}(v; S_u, S_v)} \xrightarrow{t_2} \underbrace{\{[0, 0, 0], [0, 1, 0]\}}_{I_{\text{CAW}}(w; S_u, S_v)} \xrightarrow{t_1} \underbrace{\{[0, 0, 0], [0, 0, 1]\}}_{I_{\text{CAW}}(z; S_u, S_v)}$$

and the walks associated with $(z, v, t_3)$ are (here we omit underbraces for readability):

$$\{[1, 0, 0], [0, 0, 1]\} \xrightarrow{t_1} \{[0, 1, 0], [0, 1, 0]\}$$

$$\{[0, 0, 0], [2, 0, 0]\} \xrightarrow{t_1} \{[0, 0, 0], [0, 1, 0]\}$$

$$\{[0, 0, 0], [2, 0, 0]\} \xrightarrow{t_2} \{[0, 1, 0], [0, 1, 0]\} \xrightarrow{t_1} \{[1, 0, 0], [0, 0, 1]\}$$

In this example, assume that MLPs used to encode each walk correspond to identity mappings. Then, the sum of the elements in each set is injective since each element of the sets in the anonymized walks are one-hot vectors. We note that, in this example, we can simply choose a RNN that sums the vectors in each sequence (walks), and then apply a mean readout layer (or pooling aggregator) to obtain distinct representations for $(u, v, t_3)$ and $(z, v, t_3)$.

We now use the example in Figure S3(Right) to show that MP-TGNs can separate events that CAW cannot. To see why MP-TGNs can separate the events $(u, z, t_4)$ and $(u', z, t_4)$, it suffices to observe that the 4-depth TCTs of $u$ and $u'$ are non-isomorphic. Thus, a MP-TGN with injective layers could distinguish such events. Now, let us take a look at the anonymized walks for $(u, z, t_4)$:

$$\underbrace{\{[1, 0], [0, 0]\}}_{I_{\text{CAW}}(u; S_u, S_z)} \xrightarrow{t_1} \underbrace{\{[0, 1], [0, 0]\}}_{I_{\text{CAW}}(v; S_u, S_z)}$$

$$\underbrace{\{[0, 0], [1, 0]\}}_{I_{\text{CAW}}(z; S_u, S_z)} \xrightarrow{t_1} \underbrace{\{[0, 0], [0, 1]\}}_{I_{\text{CAW}}(w; S_u, S_z)}$$

and for $(u', z, t_4)$:

$$\underbrace{\{[1,0],[0,0]\}}_{I_{\text{CAW}}(u';S_{u'},S_z)} \xrightarrow{t_1} \underbrace{\{[0,1],[0,0]\}}_{I_{\text{CAW}}(v';S_{u'},S_z)}$$

$$\underbrace{\{[0,0],[1,0]\}}_{I_{\text{CAW}}(z;S_{u'},S_z)} \xrightarrow{t_1} \underbrace{\{[0,0],[0,1]\}}_{I_{\text{CAW}}(w;S_{u'},S_z)}$$

Since the sets of walks are identical, they must have the same embedding. Therefore, there is no CAW model that can separate these two events. □

## B.8 Proof of Proposition 6: Injective MP-TGNs and the temporal WL test

We want to prove that injective MP-TGNs can separate two temporal graphs if and only if the temporal WL does the same. Our proof comprises two parts. We first show that if an MP-TGN produces different multisets of embeddings for two non-isomorphic temporal graphs $\mathcal{G}(t)$ and $\mathcal{G}'(t)$, then the temporal WL decides these graphs are not isomorphic. Then, we prove that, if the temporal WL decides $\mathcal{G}(t)$ and $\mathcal{G}'(t)$ are non-isomorphic, there is an injective MP-TGN (i.e., with injective message-passing layers) that outputs distinct multisets of embeddings.

**Statement 1:** Temporal WL is at least as powerful as MP-TGNs.

See Lemma B2 for proof.

**Statement 2:** Injective MP-TGN is at least as powerful as temporal WL.

*Proof.* To prove this, we can repurpose the proof of Theorem 3 in [43]. In particular, we assume MP-TGNs that meet the injective requirements of Proposition 2, i.e., MP-TGNs that implement injective aggregate and update functions on multisets of hidden representations from temporal neighbors. Following their footprints, we prove that there is a injection $\varphi$ to the set of embeddings of all nodes in a temporal graph from their respective colors in the temporal WL test. We do so via induction on the number of layers $\ell$. To achieve our purpose, we can assume identity memory without loss of generality.

The base case ($\ell = 0$) is straightforward since the temporal WL test initializes colors with node features. We now focus on the inductive step. Suppose the proposition holds for $\ell - 1$. Note that our update function:

$$h_v^{(\ell)}(t) = \text{UPDATE}^{(\ell)}\left(h_v^{(\ell-1)}(t), \text{AGG}^{(\ell)}(\{\!\!\{(h_u^{(\ell-1)}(t), t - t', e) \mid (u, e, t') \in \mathcal{N}(v,t)\}\!\!\})\right)$$

can be rewritten using $\varphi$ as a function of node colors:

$$h_v^{(\ell)}(t) = \text{UPDATE}^{(\ell)}\left(\varphi(c^{\ell-1}(v)), \text{AGG}^{(\ell)}(\{\!\!\{(\varphi(c^{\ell-1}(u)), t - t', e) \mid (u, e, t') \in \mathcal{N}(v,t)\}\!\!\})\right).$$

Note that the composition of injective functions is also injective. In addition, time-shifting operations are also injective. Thus we can construct an injection $\psi$ such that:

$$h_v^{(\ell)}(t) = \psi\left(c^{\ell-1}(v), \{\!\!\{(c^{\ell-1}(u), t', e) \mid (u, e, t') \in \mathcal{N}(v,t)\}\!\!\}\right)$$
$$= \psi\left(c^{\ell-1}(v), \{\!\!\{(c^{\ell-1}(u), t', e_{uv}(t')) \mid (v, u, t') \in \mathcal{G}(t)\}\!\!\}\right)$$

since there exists an element $(u, e, t') \in \mathcal{N}(v, t)$ if and only if there is an event $(u, v, t') \in \mathcal{G}(t)$ with feature $e_{uv}(t') = e$.

Then, we can write:

$$h_v^{(\ell)}(t) = \psi \circ \text{HASH}^{-1} \circ \text{HASH}\left(c^{\ell-1}(v), \{\!\!\{(c^{\ell-1}(u), t', e_{uv}(t')) \mid (u, v, t') \in \mathcal{G}(t)\}\!\!\}\right)$$
$$= \psi \circ \text{HASH}^{-1}(c^{(\ell)}(v))$$

Note $\varphi = \psi \circ \text{HASH}^{-1}$ is injective since it is a composition of two injective functions. We then conclude that if the temporal WL test outputs different multisets of colors, then a suitable MP-TGN outputs different multisets of embeddings. □

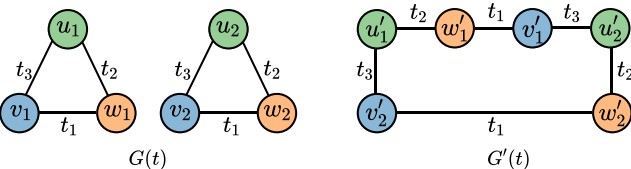

Figure S4: Examples of temporal graphs for which MP-TGNs cannot distinguish the diameter, girth, and number of cycles. For any node in $\mathcal{G}(t)$ (e.g., $u_1$), there is a corresponding one in $\mathcal{G}'(t)$ ($u_1'$) whose TCTs are isomorphic.

### B.9  Proof of Proposition 7: MP-TGNs and CAWs fail to decide some graph properties

**Statement 1:** MP-TGNs fail to decide some graph properties.

*Proof.* Adapting a construction by Garg et al. [9], we provide in Figure S4 an example that demonstrates Proposition 7. Colors denote node features, and all edge features are identical. The temporal graphs $\mathcal{G}(t)$ and $G'(t)$ are non-isomorphic and differ in properties such as diameter ($\infty$ for $\mathcal{G}(t)$ and 3 for $\mathcal{G}'(t)$), girth (3 for $\mathcal{G}(t)$ and 6 for $\mathcal{G}'(t)$), and number of cycles (2 for $\mathcal{G}(t)$ and 1 for $\mathcal{G}'(t)$). In spite of that, for $t > t_3$, the set of embeddings of nodes in $\mathcal{G}(t)$ is the same as that of nodes in $\mathcal{G}'(t)$ and, therefore, MP-TGNs cannot decide these properties. In particular, by constructing the TCTs of all nodes at time $t > t_3$, we observe that the TCTs of the pairs $(u_1, u_1')$, $(u_2, u_2')$, $(v_1, v_1')$, $(v_2, v_2')$, $(w_1, w_1')$, $(w_2, w_2')$ are isomorphic and, therefore, they can not be distinguished (Lemma 1).  □

**Statement 2:** CAWs fail to decide some graph properties.

Since CAW does not provide a recipe to obtain graph-level embeddings, we first define such a procedure. Let $\mathcal{G}(t)$ be a temporal graph given as a set of events. We sequentially compute event embeddings $h_\gamma$ for each event $\gamma = (u, v, t') \in \mathcal{G}(t)$ respecting the temporal order (two or more events at the same timestamp are computed in parallel). We then apply a readout layer to the set of event embeddings to obtain a graph-level representation. We provide a proof assuming this procedure.

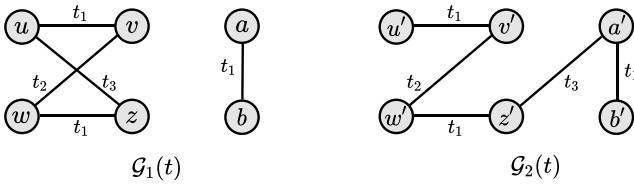

Figure S5: Examples of temporal graphs with different static properties, such as diameter, girth, and number of cycles. CAWs fail to distinguish $\mathcal{G}_1(t)$ and $\mathcal{G}_2(t)$.

*Proof.* We can adapt our construction in Figure 2 [rightmost] to extend Proposition 7 to CAW. The idea consists of creating two temporal graphs with different diameters, girths, and numbers of cycles that comprise events that CAW cannot separate — Figure S5 provides one such construction. In particular, CAW obtains identical embeddings for $(u, z, t_3)$ and $(a', z', t_3)$ (as shown in Proposition 5). The remaining events are the same up to node re-labelling and thus also lead to identical embeddings. Therefore, CAW cannot distinguish $\mathcal{G}_1(t)$ and $\mathcal{G}_2(t)$ although they clearly differ in diameter, girth, and number of cycles.  □

### B.10  Proof of Lemma 2

We now show that the $k$-th component of the relative positional features $r_{u \to v}^{(t)}$ corresponds to the number of occurrences of $u$ at the $k$-th layer of the monotone TCT of $v$, and this is valid for all pairs of nodes $u$ and $v$ of the dynamic graph. We proceed with a proof by induction.

[*Base case*] Let us consider $t = 0$, i.e., no events have occurred. By definition, $r_{u \to v}^{(0)}$ is the zero vector if $u \neq v$, indicating that node $u$ does not belong to the TCT of $v$. If $u = v$, then $r_{u \to u}^{(0)} = [1, 0, \ldots, 0]$ corresponds to count 1 for the root of the TCT of $u$. Thus, for $t = 0$, the proposition holds.

[*Induction step*] Assume that the proposition holds for all nodes and any time instant up to $t$. We will show that after the event $\gamma = (u, v, t)$ at time $t$, the proposition remains true.

Note that the event $\gamma$ only impacts the monotone TCTs of $u$ and $v$. The reason is that the monotone TCTs of all other nodes have timestamps lower than $t$, which prevents the event $\gamma$ from belonging to any path (with decreasing timestamps) from the root.

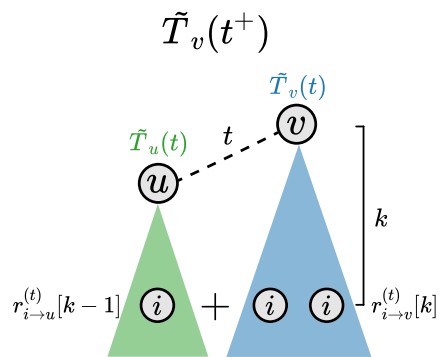

Without loss of generality, let us now consider the impact of $\gamma$ on the monotone TCT of $v$. Figure S6 shows how the TCT of $v$ changes after $\gamma$, i.e., how it goes from $\tilde{T}_v(t)$ to $\tilde{T}_v(t^+)$. In particular, the process attaches the TCT of $u$ to the root node $v$. Under this change, we need to update the counts of all nodes $i$ in $\tilde{T}_u(t)$ regarding how many times it appears in $\tilde{T}_v(t^+)$. We do so by adding the counts in $\tilde{T}_u(t)$ (i.e., $r^{(t)}_{i \to u}$) to $\tilde{T}_v(t)$ (i.e., $r^{(t)}_{i \to v}$), accounting for the 1-layer

Figure S6: Illustration of how the monotone TCT of $v$ changes after an event between $u$ and $v$ at time $t$. This allows us to see how to update the positional features of any node $i$ of the dynamic graph that belongs to $\tilde{T}_u(t)$ relative to $v$.

mismatch, since $\tilde{T}_u(t)$ is attached to the first layer. This can be easily achieved with the shift matrix $P = \begin{bmatrix} 0 & 0 \\ I_{d-1} & 0 \end{bmatrix}$ applied to the counts of any node $i$ in $\tilde{T}_u(t)$, i.e.,

$$r^{(t^+)}_{i \to v} = P\, r^{(t)}_{i \to u} + r^{(t)}_{i \to v} \qquad \forall i \in \mathcal{V}^{(t)}_u,$$

where $\mathcal{V}^{(t)}_u$ comprises the nodes of the original graph that belong to $\tilde{T}^{(t)}_u$.

Similarly, the event $\gamma$ also affects the counts of nodes in the monotone TCT of $v$ w.r.t. the monotone TCT of $u$. To account for that change, we follow the same procedure and update $r^{(t^+)}_{j \to u} = P\, r^{(t)}_{j \to v} + r^{(t)}_{j \to u}, \forall j \in \mathcal{V}^{(t)}_v$.

**Handling multiple events at the same time.** We now consider the setting where a given node $v$ interacts with multiple nodes $u_1, u_2, \ldots, u_J$ at time $t$. We can extend the computation of positional features to this setting in a straightforward manner by noting that each event leads to an independent branch in the TCT of $v$. Therefore, the update of the positional features with respect to $v$ is given by

$$r^{(t^+)}_{i \to v} = P \sum_{j=1}^{J} r^{(t)}_{i \to u_j} + r^{(t)}_{i \to v} \qquad \forall i \in \bigcup_{j=1}^{J} \mathcal{V}^{(t)}_{u_j}$$

$$\mathcal{V}^{(t^+)}_v = \mathcal{V}^{(t)}_v \bigcup_{j=1}^{J} \mathcal{V}^{(t)}_{u_j}.$$

We note that the updates of the positional features of $u_1, \ldots, u_J$ remain untouched if they do not interact with other nodes at time $t$.

### B.11 Proof of Proposition 8: Injective function on temporal neighborhood

*Proof.* To capture the intuition behind the proof, first consider a multiset $M$ such that $|M| < 4$. We can assign a unique number $\psi(m) \in \{1, 2, 3, 4\}$ to any distinct element $m \in M$. Also, the function $h(m) = 10^{-\psi(m)}$ denotes the decimal expansion of $\psi(m)$ and corresponds to reserving one decimal place for each unique element $m \in M$. Since there are less than 10 elements in the multiset, note that $\sum_m h(m)$ is unique for any multiset $M$.

To prove the proposition, we also leverage the well-known fact that the Cartesian product of two countable sets is countable — the Cantor's (bijective) pairing function $z : \mathbb{N} \times \mathbb{N} \to \mathbb{N}$, with $z(n_1, n_2) = \frac{(n_1 + n_2)(n_1 + n_2 + 1)}{2} + n_2$, provides a proof for that.

Here, we consider multisets $M = \{\!\{(x_i, e_i, t_i)\}\!\}$ whose tuples take values on the Cartesian product of the countable sets $\mathcal{X}$, $\mathcal{E}$, and $\mathcal{T}$ — the latter is also assumed to be bounded. In addition, we assume

the lengths of all multisets are bounded by $N$, i.e., $|M| < N$ for all $M$. Since $\mathcal{X}$ and $\mathcal{E}$ are countable, there exists an enumeration function $\psi : \mathcal{X} \times \mathcal{E} \to \mathbb{N}$ for all $M$. Without loss of generality, we assume $\mathcal{T} = \{1, 2, t_{\max}\}$. We want to show that exists a function of the form $\sum_i 10^{-k\psi(x_i, e_i)} \alpha^{-\beta t_i}$ that is unique on any multiset $M$.

Our idea is to separate a range of $k$ decimal slots for each unique element $(x_i, e_i, \cdot)$ in the multiset. Each such a range has to accommodate at least $t_{\max}$ decimal slots (one for each value of $t_i$). Finally, we need to make sure we can add up to $N$ values at each decimal slot.

Formally, we map each tuple $(x_i, e_i, \cdot)$ to one of $k$ decimal slots starting from $10^{-k\psi(x_i, e_i)}$. In particular, for each element $(x_i, e_i, t_i = j)$ we add one unit at the $j$-th decimal slot after $10^{-k\psi(x_i, e_i)}$. Also, to ensure the counts for $(x_i, e_i, j)$ and $(x_i, e_i, l \neq j)$ do not overlap, we set $\beta = \lceil \log_{10} N \rceil$ since no tuple can repeat more than $N$ times. We use $\alpha = 10$ as we shift decimals. Finally, to guarantee that each range encompasses $t_{\max}$ slots of $\beta$ decimals, we set $k = \beta(t_{\max} + 1)$. Therefore, the function

$$\sum_i 10^{-k\psi(x_i, e_i)} \alpha^{-\beta t_i}$$

is unique on any multiset $M$. We note that, without loss of generality, one could choose a different basis (other than 10). $\qquad \square$

## B.12 Proof of Proposition 9: Expressiveness of PINT: link prediction

*Proof.* We now show that PINT (with relative positional features) is strictly more powerful than MP-TGN and CAW in distinguishing edges of temporal graphs. Leveraging Proposition 5, it suffices to show that PINT is at least as powerful as both CAW and MP-TGN.

**Statement 1:** PINT is at least as powerful as MP-TGNs.

Since PINT is a generalization of MP-TGNs with injective aggregation/update layers, it derives that it is at least as powerful as MP-TGNs. We can set the model's parameters associated with positional features to zero and obtain an equivalent MP-TGN.

**Statement 2:** PINT is at least as powerful as CAW.

We wish to show that for any pair of events that PINT cannot distinguish, CAW also cannot distinguish it. Let us consider the events $(u, v, t)$ and $(u', v', t)$ of a temporal graph. Formally, we want to prove that if $\{\!\{T_u(t), T_v(t)\}\!\} = \{\!\{T_{u'}(t), T_{v'}(t)\}\!\}$ (i.e., the multisets contain TCTs that are pairwise isomorphic), then $\{\!\{\text{ENC}(W; S_u, S_v)\}\!\}_{W \in \{S_u \cup S_v\}} = \{\!\{\text{ENC}(W'; S_{u'}, S_{v'})\}\!\}_{W' \in \{S_{u'} \cup S_{v'}\}}$, where ENC denotes the walk-encoding function of CAW. Importantly, for the sake of this proof, we assume that all TCTs here are augmented with positional features, characterizing edge embeddings obtained from PINT.

Without loss of generality, we can assume that $T_u(t) \cong T_{u'}(t)$ and $T_v(t) \cong T_{v'}(t)$. By Lemma B1, we know that the corresponding monotone TCTs are also isomorphic: $\tilde{T}_u(t) \cong \tilde{T}_{u'}(t)$, and $\tilde{T}_v(t) \cong \tilde{T}_{v'}(t)$ with associated bijections $f_1 : V(\tilde{T}_u(t)) \to V(\tilde{T}_{u'}(t))$ and $f_2 : V(\tilde{T}_v(t)) \to V(\tilde{T}_{v'}(t))$.

We can construct a tree $T_{uv}$ by attaching $\tilde{T}_u(t)$ and $\tilde{T}_v(t)$ to a (virtual) root node $uv$ — without loss of generality, $u$ and $v$ are the left-hand and right-hand child of $uv$, respectively. We can follow the same procedure and create the tree $T_{u'v'}$ by attaching the TCTs $\tilde{T}_{u'}(t)$ and $\tilde{T}_{v'}(t)$ to a root node $u'v'$. Since the left-hand and right-hand subtrees of $T_{uv}$ and $T_{u'v'}$ are isomorphic, then $T_{uv}$ and $T_{u'v'}$ are also isomorphic. Let $f : V(T_{uv}) \to V(T_{u'v'})$ denote the bijection associated with the augmented trees. We also assume $f$ is constructed by preserving the bijections $f_1$ and $f_2$ defined between the original monotone TCTs: this ensures that $f$ does not map any node in $V(\tilde{T}_u(t))$ to a node in $V(\tilde{T}_{v'}(t))$, for instance. We have that

$$[r^{(t)}_{\sharp i \to u} \| r^{(t)}_{\sharp i \to v}] = [r^{(t)}_{\sharp f(i) \to u'} \| r^{(t)}_{\sharp f(i) \to v'}] \quad \forall i \in V(T_{uv}) \setminus \{uv\}$$

Note that we use the function $\sharp$ (that maps nodes in the TCT to nodes in the dynamic graph) here because the positional feature vectors are defined for nodes in the dynamic graph.

To guarantee that two encoded walks are identical $\text{ENC}(W; S_u, S_v) = \text{ENC}(W'; S_{u'}, S_{v'})$, it suffices to show that the anonymized walks are equal. Thus, we turn our problem into showing that for any walk $W = (w_0, t_0, w_1, t_1, \dots)$ in $S_u \cup S_v$, there exists a corresponding one

$W' = (w'_0, t_0, w'_1, t_1, \dots)$ in $S_{u'} \cup S_{v'}$ such that $I_{\mathrm{CAW}}(w_i; S_u, S_v) = I_{\mathrm{CAW}}(w'_i; S_{u'}, S_{v'})$ for all $i$. Recall that $I_{\mathrm{CAW}}(w_i; S_u, S_v) = \{g(w_i; S_u), g(w_i; S_v)\}$, where $g(w_i; S_u)$ is a vector whose $k$-component stores how many times $w_i$ appears in a walk from $S_u$ at position $k$.

A key observation is that there is an equivalence between deanonymized root-leaf paths in $T_{uv}$ and walks in $S_u \cup S_v$ (disregarding the virtual root node). By deanonymized, we mean paths where node identities (in the temporal graph) are revealed by applying the function $\sharp$. Using this equivalence, it suffices to show that

$$g(\sharp i; S_u) = g(\sharp f(i); S_{u'}) \text{ and } g(\sharp i; S_v) = g(\sharp f(i); S_{v'}) \quad \forall i \in V(T_{uv}) \setminus \{uv\}$$

Suppose there is an $i \in V(T_{uv}) \setminus \{uv\}$ such that $g(\sharp i; S_u) \neq g(\sharp f(i); S_{u'})$. Without loss of generality, suppose this holds for the $\ell$-th entry of the vectors.

We know there are exactly $r^{(t)}_{a \to u}[\ell]$ nodes at the $\ell$-th level of $\tilde{T}_u(t)$ that are associated with $a = \sharp i \in V(\mathcal{G}(t))$. We denote by $\Psi$ the set comprising such nodes. It also follows that computing $g(\sharp i; S_u)[\ell]$ is the same as summing up the amount of leaves of each subtree of $\tilde{T}_u(t)$ rooted at $\psi \in \Psi$, which we denote as $\mathscr{D}(\psi; \tilde{T}_u(t))$, i.e.,

$$g(\sharp i; S_u)[\ell] = \sum_{\psi \in \Psi} \mathscr{D}(\psi; \tilde{T}_u(t)).$$

Since we assume $g(\sharp i; S_u)[\ell] \neq g(\sharp f(i); S_{u'})[\ell]$, then it holds that

$$g(\sharp i; S_u)[\ell] \neq g(\sharp f(i); S_{u'})[\ell] \Rightarrow \sum_{\psi \in \Psi} \mathscr{D}(\psi; \tilde{T}_u(t)) \neq \sum_{\psi \in \Psi} \mathscr{D}(f(\psi); \tilde{T}_{u'}(t)) \tag{S5}$$

Note that the subtree of $\tilde{T}_u$ rooted at $\psi$ should be isomorphic to the subtree of $\tilde{T}_{u'}$ rooted at $f(\psi)$, and therefore have the same number of leaves. However, the RHS of Equation S5 above implies there is a $\psi \in \Psi$ for which $\mathscr{D}(\psi; \tilde{T}_u) \neq \mathscr{D}(f(\psi); \tilde{T}_{u'})$, reaching a contradiction. The same argument can be applied to $v$ and $v'$ to prove that $g(\sharp i; S_v) = g(\sharp f(i); S_{v'})$. $\qquad \square$

## C Additional related works

**Structural features for static GNNs.** Using structural features to enhance the power of GNNs is an active research topic. Bouritsas et al. [48] improved GNN expressivity by incorporating counts of local structures in the message-passing procedure, e.g, the number triangles a node appears on. These counts depend on identifying subgraph isomorphisms and, naturally, can become intractable depending on the chosen substructure. Li et al. [52] proposed increasing the power of GNNs using distance encodings, i.e., augmenting original node features with distance-based ones. In particular, they compute the distance between a node set whose representation is to be learned and each node in the graph. To alleviate the cost of distance encoding, an alternative is to learn absolute position encoding schemes [51, 61, 62], that try to summarize the role each node plays in the overall graph topology. We note that another class of methods uses random features to boost the power of GNNs [47, 58]. However, these models are referred to be hard to converge and obtain noisy predictions [62].

The most trivial difference between these approaches and PINT is that our relative positional features account for temporal information. On a deeper level, our features summarize the role each node plays in each other's monotone TCT instead of measuring, e.g., pair-wise distances in the original graph or counting substructures. Also, our scheme leverages the temporal aspect to achieve computational tractability, updating features incrementally as events unroll. Finally, while some works proposing structural features for static GNNs present marginal gains [62], PINT exhibits significant performance gains in real-world temporal link prediction tasks.

**Other models for temporal graphs.** Representation learning for dynamic graphs is a broad and diverse field. In fact, strategies to cope with the challenge of modeling dynamic graphs can come in many flavors, including simple aggregation schemes [18], walk-aggregating methods [53], and combinations of sequence models with GNNs [56, 59]. For instance, Seo et al. [59] used a spectral graph convolutional network [49] to encode graph snapshots followed by a graph-level LSTM [13]. Manessi et al. [55] followed a similar approach but employed a node-level LSTM, with parameters shared across the nodes. Sankar et al. [57] proposed a fully attentive model based on graph attention

networks [34]. Pareja et al. [56] applied a recurrent neural net to dynamically update the parameters of a GCN. Gao and Ribeiro [50] compared the expressive power of two classes of models for discrete dynamic graphs: time-and-graph and time-then-graph. The former represents the standard approach of interleaving GNNs and sequence (e.g., RNN) models. In the latter class, the models first capture node and edge dynamics using RNNs, and are then feed into graph neural networks. The authors showed that time-then-graph has expressive advantage over time-and-graph approaches under mild assumptions. For an in-depth review of representation learning for dynamic graphs, we refer to the survey works by Kazemi et al. [14] and Skarding et al. [60].

While most of the early works focused on discrete-time dynamic graphs, we have recently witnessed a rise in interest in models for event-based temporal graphs (i.e., CTDGs). The reason is that models for DTDGs may fail to leverage fine-grained temporal and structural information that can be crucial in many applications. In addition, it is hard to specify meaningful time intervals for different tasks. Thus, modern methods for temporal graphs explicitly incorporate timestamp information into sequence/graph models, achieving significant performance gains over approaches for DTDGs [38]. Appendix A provides a more detailed presentation of CAW, TGN-Att, and TGAT, which are among the best performing models for link prediction on temporal graphs. Besides these methods, JODIE [16] applies two RNNs (for the source and target nodes of an event) with a time-dependent embedding projection to learn node representations of item-user interaction networks. Trivedi et al. [32] employed RNNs with a temporally attentive module to update node representations. APAN [63] consists of a memory-based TGN that uses attention mechanism to update memory states using multi-hop temporal neighborhood information. Makarov et al. [54] proposed incorporating edge embeddings obtained from CAW into MP-TGNs' memory and message-passing computations.

# D Datasets and implementation details

## D.1 Datasets

In our empirical evaluation, we have considered six datasets for dynamic link prediction: Reddit[1], Wikipedia[2], UCI[3], LastFM[4], Enron[5], and Twitter. Reddit is a network of posts made by users on subreddits, considering the 1,000 most active subreddits and the 10,000 most active users. Wikipedia comprises edits made on the 1,000 most edited Wikipedia pages by editors with at least 5 edits. Both Reddit and Wikipedia networks include links collected over one month, and text is used as edge features, providing informative context. The LastFM dataset is a network of interactions between user and the songs they listened to. UCI comprises students' posts to a forum at the University of California Irvine. Enron contains a collection of email events between employees of the Enron Corporation, before its bankruptcy. The Twitter dataset is a non-bipartite net where nodes are users and interactions are retweets. Since Twitter is not publicly available, we build our own version by following the guidelines by Rossi et al. [27]. We use the data available from the 2021 Twitter RecSys Challenge and select 10,000 nodes and their associated interactions based on node participation: number of interactions the node participates in. We also apply multilingual BERT to obtain text representations of retweets (edge features).

Table S1 reports statistics of the datasets such as number of temporal nodes and links, and the dimensionality of the edge features. We note that UCI, Enron, and LastFM represent non-attributed networks and therefore do not contain feature vectors associated with the events. Also, the node features for all datasets are vectors of zeros [42].

Table S1: Summary statistics of the datasets.

| Dataset | #Nodes | #Events | #Edge feat. | Bipartite? |
|---|---|---|---|---|
| Reddit | 10,984 (10,000 / 984) | 672,447 | 172 | Yes |
| Wikipedia | 9,227 (8,227 / 1,000) | 157,474 | 172 | Yes |
| Twitter | 8,925 | 406,564 | 768 | No |
| UCI | 1,899 | 59,835 | - | No |
| Enron | 184 | 125,235 | - | No |
| LastFM | 1,980 (980 / 1,000) | 1,293,103 | - | Yes |

## D.2 Implementation details

We train all models in link prediction tasks in a self-supervised approach. During training, we generate negative samples: for each actual event $(z, w, t)$ (class 1), we create a fake one $(z, w', t)$ (class 0) where $w'$ is uniformly sampled from the set of nodes, and both events have the same edge feature vector.

To ensure a fair comparison, we mainly rely on the original repositories and guidelines. For instance, regarding the training of MP-TGNs (including PINT), we mostly follow the setup and choices in the implementation available in [27]. In particular, we apply the Adam optimizer with learning rate $10^{-4}$ during 50 epochs with early stopping if there is no improvement in validation AP for 5 epochs. In addition, we use batch size 200 for all methods. We report statistics (mean and standard deviation) of the performance metric (AP) over ten runs.

**MP-TGNs.** For TGN-Att, we follow Rossi et al. [27] and sample either ten or twenty temporal neighbors with memory dimensionality equal to 32 (Enron), 100 (UCI, Twitter), or 172 (Reddit, Wikipedia, LastFM), node embedding dimension equal to 100, and two attention heads. We use a memory unit implemented as a GRU, and update the state of each node based on only its most recent message. For TGAT, we use twenty temporal neighbors and two layers.

**CAW.** We conduct model selection using grid search over: i) time decay $\alpha \in \{0.01, 0.1, 0.25, 0.5, 1.0, 2.0, 4.0, 10.0, 100.0\} \times 10^{-6}$, ii) number of walks $M \in \{1, 2, 3, 4, 5\}$;

---

[1] http://snap.stanford.edu/jodie/reddit.csv
[2] http://snap.stanford.edu/jodie/wikipedia.csv
[3] http://konect.cc/networks/opsahl-ucforum/
[4] http://snap.stanford.edu/jodie/lastfm.csv
[5] https://www.cs.cmu.edu/~./enron/

and iii) walk length $L \in \{32, 64, 128\}$. The best combination of hyperparameters is shown in Table S2. The remaining training choices follows the default values from the original implementation. Importantly, we note that TGN-Att's original evaluation setup is different from CAW's. Thus, we adapted CAW's original repo to reflect these differences and ensure a valid comparison.

Table S2: Optimal hyperparameters for CAW.

| Dataset | Time decay $\alpha$ | #Walks | Walk length |
|---------|---------------------|--------|-------------|
| Reddit | $10^{-8}$ | 32 | 3 |
| Wikipedia | $4 \times 10^{-6}$ | 64 | 4 |
| Twitter | $10^{-6}$ | 64 | 3 |
| UCI | $10^{-5}$ | 64 | 2 |
| Enron | $10^{-6}$ | 64 | 5 |
| LastFM | $10^{-6}$ | 64 | 2 |

**PINT.** We use $\alpha = 2$ (in the exponential aggregation function), and experiment with learned and fixed $\beta$. We apply a relu function to avoid negative values of $\beta$, which could lead to unstable training. We do grid search as the follow: when learning beta, we consider initial values for $\beta \in \{0.1, 0.5\}$; for the fixed case (`requires_grad=False`), we evaluate $\beta \in \{10^{-3} \times |\mathcal{N}|, 10^{-4} \times |\mathcal{N}|, 10^{-5} \times |\mathcal{N}|\}$ — $|\mathcal{N}|$ denotes the number of temporal neighbors — and always apply memory as in the original implementation of TGN-Att. We consider number message passing layers $\ell$ in $\{1, 2\}$. Also, we apply neighborhood sampling with the number of neighbors in $\{10, 20\}$, and update the state of a node based on its most recent message. We then carry out model selection based on AP values obtained during validation. Overall, the models with fixed $\beta$ led to better results. Table S3 reports the optimal hyperparameters for PINT found via automatic model selection.

In all experiments, we use relative positional features with $d = 4$ dimensions. For computational efficiency, we update the relative positional features only after processing a batch, factoring in all events from that batch. Note that this prevents information linkage as these positional features take effect after prediction. In addition, since temporal events repeat (in the same order) at each epoch, we also speed up PINT's training procedure by precomputing and saving the positional features for each batch. To save up space, we store the positional features as sparse matrices.

Table S3: Optimal hyperparameters for PINT.

| Dataset | $\beta/|\mathcal{N}|$ | #Neighbors ($|\mathcal{N}|$) | #Layers |
|---------|------------------------|------------------------------|---------|
| Reddit | $10^{-5}$ | 10 | 2 |
| Wikipedia | $10^{-4}$ | 10 | 2 |
| Twitter | $10^{-5}$ | 20 | 2 |
| UCI | $10^{-5}$ | 10 | 2 |
| Enron | $10^{-5}$ | 20 | 2 |
| LastFM | $10^{-4}$ | 10 | 2 |

**Hardware.** For all experiments, we use Tesla V100 GPU cards and consider a memory budget of 32GB of RAM.

# E   Deletion and node-level events

Rossi et al. [27] propose handling edge deletions by simply updating memory states of the edge's endpoints, as if we were dealing with a usual edge addition. However, it is not discussed whether the edge in question should be excluded from the event list or if we should just add a novel event with edge features that characterize deletion. If we choose the former, we may be unable to recover the memory state of a node from its monotone TCT and the original node features. Removing an edge from the event list also affects the computation of node embeddings. Therefore, we advise practitioners to do the latter when using PINT. It is worth mentioning that the vast majority of models for temporal interaction prediction do not consider the possibility of deletion events.

Regarding node-level events, PINT can accommodate node addition by simply creating novel memory states. To deal with node feature updates, we can create an edge event with both endpoints on that

node, inducing a self-loop in the dynamic graph. Also, we can combine (e.g., concatenate) the temporal features in message-passage operations, similarly to the general formulation of the MP-TGN framework [27]. Finally, we can deal with the removal of a node $v$ by following our previous (edge deletion) procedure to delete all edges with endpoints in $v$.

## F Additional experiments

**Time comparison.** Figure S7 compares the time per epoch for PINT and for the prior art (CAW, TGN-Att, and TGAT) in the Enron and LastFM datasets. Following the trend in Figure 7, Figure S7 further supports that PINT is generally slower than other MP-TGNs but, after a few training epochs, is orders of magnitude faster than CAW. In the case of Enron, the time CAW takes to complete an epoch is much higher than the time we need to preprocess PINT's positional features.

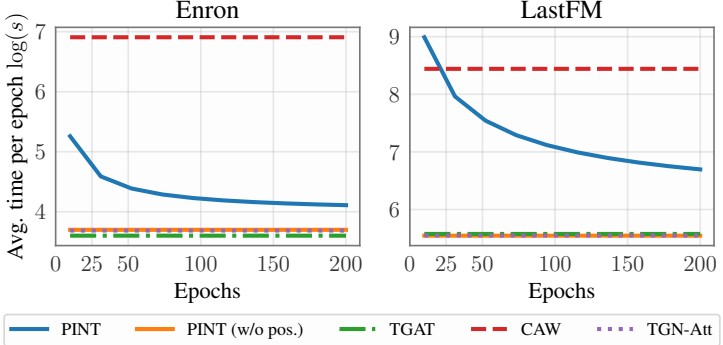

Figure S7: Time comparison: PINT versus TGNs (in log-scale) on Enron and LastFM.

**Experiments on node classification.** For completeness, we also evaluate PINT on node-level tasks (Wikipedia and Reddit). We follow closely the experimental setup in Rossi et al. [27] and compare against the baselines therein. Table S4 shows that PINT ranks first on Reddit and second on Wikipedia. The values for PINT reflect the outcome of 5 repetitions.

Table S4: Results for node classification (AUC).

|  | **Wikipedia** | **Reddit** |
|---|---|---|
| CTDNE | $75.89 \pm 0.5$ | $59.43 \pm 0.6$ |
| JODIE | $84.84 \pm 1.2$ | $61.83 \pm 2.7$ |
| TGAT | $83.69 \pm 0.7$ | $65.56 \pm 0.7$ |
| DyRep | $84.59 \pm 2.2$ | $62.91 \pm 2.4$ |
| TGN-Att | $87.81 \pm 0.3$ | $67.06 \pm 0.9$ |
| PINT | $87.59 \pm 0.6$ | $67.31 \pm 0.2$ |

## Supplementary References

[47] R. Abboud, I. I. Ceylan, M. Grohe, and T. Lukasiewicz. The surprising power of graph neural networks with random node initialization. In *International Joint Conference on Artificial Intelligence (IJCAI)*, 2021.

[48] G. Bouritsas, F. Frasca, S. Zafeiriou, and M. M. Bronstein. Improving graph neural network expressivity via subgraph isomorphism counting. In *Arxiv e-prints*, 2020.

[49] M. Defferrard, X. Bresson, and P. Vandergheynst. Convolutional neural networks on graphs with fast localized spectral filtering. In *Advances in Neural Information Processing Systems (NeurIPS)*, 2018.

[50] J. Gao and B. Ribeiro. On the equivalence between temporal and static graph representations for observational predictions. *ArXiv*, 2103.07016, 2021.

[51] D. Kreuzer, D. Beaini, W. L. Hamilton, V. Letourneau, and P. Tossou. Rethinking graph transformers with spectral attention. In *Advances in Neural Information Processing Systems (NeurIPS)*, 2021.

[52] P. Li, Y. Wang, H. Wang, and J. Leskovec. Distance encoding: Design provably more powerful neural networks for graph representation learning. In *Advances in Neural Information Processing Systems (NeurIPS)*, 2020.

[53] S. Mahdavi, S. Khoshraftar, and A. An. dynnode2vec: Scalable dynamic network embedding. In *International Conference on Big Data*, 2018.

[54] I. Makarov, A. V. Savchenko, A. Korovko, L. Sherstyuk, N. Severin, A. Mikheev, and D. Babaev. Temporal graph network embedding with causal anonymous walks representations. *ArXiv*, 2108.08754, 2021.

[55] F. Manessi, A. Rozza, and M. Manzo. Dynamic graph convolutional networks. *Pattern Recognition*, 97, 2020.

[56] A. Pareja, G. Domeniconi, J. Chen, T. Ma, H. Kanezashi T. Suzumura, T. Kaler, T. B. Schardl, and C. E. Leiserson. EvolveGCN: Evolving graph convolutional networks for dynamic graphs. In *AAAI Conference on Artificial Intelligence (AAAI)*, 2020.

[57] A. Sankar, Y. Wu, L. Gou, W. Zhang, and H. Yang. DySAT: Deep neural representation learning on dynamic graphs via self-attention networks. In *International Conference on Web Search and Data Mining (WSDM)*, 2020.

[58] R. Sato, M. Yamada, and H. Kashima. Random features strengthen graph neural networks. In *SIAM International Conference on Data Mining (SDM)*, 2021.

[59] Y. Seo, M. Defferrard, P. Vandergheynst, and X. Bresson. Structured sequence modeling with graph convolutional recurrent networks. In *International Conference on Neural Information Processing (ICONIP)*, 2018.

[60] J. Skarding, B. Gabrys, and K. Musial. Foundations and modeling of dynamic networks using dynamic graph neural networks: A survey. *IEEE Access*, 9:79143–79168, 2021.

[61] B. Srinivasan and B. Ribeiro. On the equivalence between positional node embeddings and structural graph representations. In *International Conference on Learning Representations (ICLR)*, 2020.

[62] H. Wang, H. Yin, M. Zhang, and P. Li. Equivariant and stable positional encoding for more powerful graph neural networks. In *International Conference on Learning Representations (ICLR)*, 2022.

[63] X. Wang, D. Lyu, M. Li, Y. Xia, Q. Yang, X. Wang, X. Wang, P. Cui, Y. Yang, B. Sun, and Z. Guo. APAN: Asynchronous propagation attention network for real-time temporal graph embedding. *International Conference on Management of Data*, 2021.