# OpenReview forum: "Provably expressive temporal graph networks"
_NeurIPS.cc/2022/Conference — NeurIPS 2022 Accept_

### Official Review · Reviewer_HPYE · 2022-07-03

**Rating:** 5
**Confidence:** 4
**Soundness:** 3 good
**Presentation:** 2 fair
**Contribution:** 3 good

**Summary:**

This work studies the expressivity limitation of temporal graph neural networks, mainly on two state-of-the-art (SOTA) methods, TGNs (MP-TGN in paper) and CAWs, then proposes a more expressive framework, PINE.

**Questions:**

1. Around CTDG definition (line 85), it will be better to use different symbols for G since it will have two meanings before and after line 85. Before the line, G refers to simple graph (no dup edges, static or happening in specific time range); but after the line, G refers to multi-graph (have multi edges linking same nodes) whose edges are additionally distinguished by extra timestamps. It is important to highlight this shifting to avoid confusion in later content.
2. I am not sure the correctness of uniformly spaced countable set in Proposition 1 (line 93), please clarify: To transform CTDG to DTDG, we only need to put all timestamps as a real value feature on nodes or edges, and let different timestamp generate different snapshot, thus just countable will be enough. And I think this requirement may even be meaningless since even for CTDG, you can not store infinite timestamps, and with finite timestamps, CTDG and DTDG are same.
3. Need clarification of memory aggregation for equation (3) (line 114) and experiments. Equation (3) shows that you are using most recent memory aggregation in the paper, but in appendix, you investigate mean aggregation of TGN, which one are you really using? Besides, if you are using mean memory aggregation, shouldn't you decrease MP-TGN by 1 layer for fairness? Since it is a kind of graph convolution.
4. The TGN limitation example (figure 2) is unfair since TGN-Att is just a specific hyperparameter setting of original TGN paper, and I think that another setting TGN-sum can solve this counterexample, thus this is not a real counterexample for TGN.
5. TCT node symbol i_u is bad and causes confusion. In line 142, i_u should not have u as subscript since it implies a unique TCT node is assigned with u but we can have multi nodes in the same TCT layer corresponding to the same u (but different times). A better notation is required to make it clear that a real node u can appear multiple times in both the same and different layers in TCT.
6. Besides, should mapping #v also have t as an argument since it is reverse mapping in a sub tree of root (v, t)?
7. Is line 155 of definition 1 indeed s_u = s'_{f(u)}? I don't think current statement makes any sense. Omitting root node makes the notation confusing.
8. Can proposition 6 and 7 extend to random walk methods, e.g. CAW? Since without such extensions, it looks same as conclusion of reference [49] that with proper injective mappings, we can always distinguish temporal graph isomorphism like a static graph, and even the most power methods are not good for learning relations? If so, you can move this part to appendix and clarify other parts better or put other parts there, e.g., how can your method deal with deletion.
9. What is the square root term in equation (4)? It is different from proposition 8, and no explanation or intuition is provided for such modification in the paper.
10. It will be better to add more space between P and r in equation (9) and (10) since it looks like Pr on the first sight.
11. There is a highly related work CAW-TGN [1] that use CAW to get positional info for TGN which is similar to your paper, and you should add this into your comparison baseline and do ablation study.
12. Why some datasets do not have static method performance?

Minor typos:
- Figure 2: (v, z, t3) to (u, z, t3)

[1] Makarov, I., et al. (2021). Temporal graph network embedding with causal anonymous walks representations.

**Limitations:**

This paper provides the limitation of PINE which reveals the direction for future efforts and reports societal impact in appendix.

**Strengths And Weaknesses:**

The strength of this paper is:
1. Propose a concrete failure case of SOTA methods, e.g., CAW, which can be used as a benchmark for temporal graphs.
2. Studying temporal graph neural networks expressivity by temporal computation tree (TCT) and proposed temporal WL test variant.
3. Design more expressive PINE which achieve empirically better performance in 5/6 widely used datasets.

---

> ### Author Response · Authors · 2022-08-02
> **Response to Reviewer HPYE (Part 1/2)**
>
> Thanks for your helpful and insightful review. We will update the manuscript to sharpen our notation and the statement in proposition 1. Thanks to this review, we will also extend the result in proposition 7 to CAW. We hope our answers have addressed the points you have raised and improved your view of the manuscript.
>
> **Q1: "Around CTDG definition (line 85), it will be better to use different symbols for G"**
>
> Thanks for catching this. We have fixed it by using $\mathsf{G}$ to denote timestamped multi-graphs.
>
> **Q2: "I am not sure the correctness of uniformly spaced countable set in Proposition 1 (line 93), please clarify: To transform CTDG to DTDG, we only need to put all timestamps as a real value feature on nodes or edges, and let different timestamp generate different snapshot"**
>
> We agree with your construction. Nonetheless, we note our proof holds even if we do not allow edge/node features. In that case, we may need other assumptions --- such as ours. Note also that we have been careful to make the uniformly spaced condition only sufficient (not necessary). We will adapt the text to account for this discussion and sharpen proposition 1.
>
> **Q3: "Need clarification of memory aggregation for equation (3) (line 114) and experiments. Equation (3) shows that you are using most recent memory aggregation...which one are you really using?"**
>
> Thanks for the chance to clarify this. Eq. (3) only shows the memory update for node $u$ after the event $(u, v, t)$, and we assume $u$ doesn't interact with any other nodes at $t$ (lines 112-113), eliminating the need for message aggregation. Given this assumption, the update in Eq. (3) doesn't imply most-recent message aggregation. However, since we assume that two events belong to the same batch only if they occur at the same timestamp, most-recent aggregation would result in ambiguity (multiple most-recent events). That is the reason why we consider mean message aggregation in the analysis section.
>
> In Table 1, models that make use of memory (i.e., PINT and TGN-Att) employ most-recent message aggregation in batches of size 200, following the original TGN paper. We will make this clear in the revised manuscript.
>
> **Q4: "The TGN limitation example (figure 2) is unfair since TGN-Att is just a specific hyperparameter setting of original TGN paper, and I think that another setting TGN-sum can solve this counterexample"**
>
> Our point in Figure 2 [Left] (and Proposition 4) is to show that the best performing MP-TGNs (i.e., TGN-Att, TGAT) fail to distinguish nodes of very simple temporal graphs. We do not claim that MP-TGNs with other aggregation functions can not handle this example. In fact, regarding general MP-TGNs, our claim is that nodes with isomorphic TCTs cannot be distinguished, and we show an example in Figure 2 (rightmost) where the TCTs of $u$ and $z$ are isomorphic.
>
> **Q5: "TCT node symbol $i_u$ is bad and causes confusion"**
>
> Thanks for your comment. We will update the notation in the revised paper by dropping the dependence on $u$.
>
> **Q6: "should mapping $\sharp v$ also have $t$ as an argument since it is reverse mapping in a sub tree of root $(v, t)$"**
>
> Indeed, $\sharp$ depends directly on $t$. However, since we are never comparing TCTs rooted at different timestamps, we omit $\sharp $'s dependence on $t$ for a cleaner notation. We will emphasize this in the revised paper.
>
> **Q7: "Is line 155 of definition 1 indeed $s_u = s'_{f(u)}$?"**
>
> In line 145, we abbreviate the state of each node $i$ in a given TCT $T_v(t)$ by $s_i = s_{\sharp_v i}(t)$.
>
> Consequently, in Definition 1, we have that $s_u=s_{\sharp_z u}(t)$ and $s_{f(u)}=s_{\\sharp_{z^\prime} f(u)}(t)$ --- note that $u$ refers to node ids in the TCT (not the original graph).
>
> **Q8: "Can proposition 6 and 7 extend to random walk methods, e.g. CAW?"**
>
> This is an interesting question. We note that CAW does not provide a recipe to obtain node/graph embeddings but only event embeddings. Let us assume that CAW distinguishes two temporal graphs $G_1(t)$ and $G_2(t)$ (given as sets of events) at time $t$ by computing event embeddings for each element of $G_1(t)$ and $G_2(t)$ and then sending them through a readout layer. We also assume an injective readout layer -- to ensure that two graphs are distinguished if their multisets of event embeddings are different.
>
> Then, we can adapt our construction in Figure 3 [right] to extend proposition 7 to CAW. In particular, consider a temporal graph $G_1$ that includes event $(u, z, t_3)$ (and do not include $(u^\prime, z, t_3)$). Also, consider $G_2$ to include $(u^\prime, z, t_3)$ but not $(u, z, t_3)$. Thus, $G_1$ and $G_2$ only differ in these events. However, CAW assigns identical embeddings for these events (as shown in proposition 4). Therefore, CAW cannot distinguish $G_1$ and $G_2$. Finally, we note that $G_1$ and $G_2$ differ in diameter, girth, and number of cycles under an aggregated view of the temporal graphs. We will include this in the final manuscript.

---

> > ### Comment · Reviewer_HPYE · 2022-08-06
> > **Further response to Q4**
> >
> > Thank you for your reply.
> > I understand that the target of related materials is to show that some MP-TGNs which empirically reported to be sate-of-the-art have limitations.
> > But, I still think that it will be better to clarify that it is the limitation of MP-TGNs with specific memory unit as the initial layer input.
> > Since from my view, PINT is similar to MP-TGNs (equation 4 is an instance of equation 1, and equation 5 is an instance of instance 2), the only difference is PINT proposed a new initial layer input with positional-encoding.
> > Besides, in response to gAwT, you show that applying your positional-encoding to TGN-Att can give a boost which also shows the limitation comes mostly from initial layer input.
> > You propose your method as position-encoding injective temporal graph net, which makes me think that the improvement comes from changing from message-passing to position-encoding injective, while indeed the improvement comes from changing from memory unit to position-encoding injective.
> >
> > **Thus I think it is necessary to highlight that MP-TGN in all theorems refer to those specific memory unit design, otherwise, TGN-Att+PE, as an example, is an instance under your MP-TGN definition, but will overcome the issue at least in figure 2.**
> >
> > I do not have future issues except for MP-TGN definition and limitation.

---

> > > ### Author Response · Authors · 2022-08-07
> > > **RE: Further response to Q4**
> > >
> > > Thanks for your feedback. We truly appreciate your engagement in the discussion.
> > >
> > > **(Re Proposition 4) "it will be better to clarify that it is the limitation of MP-TGNs with specific memory unit as the initial layer input"**
> > >
> > > Thanks for your suggestion. We will make it explicit that Proposition 4 assumes TGN-Att uses a mean aggregator in its memory updates. The new statement of Proposition 4 now reads as:
> > >
> > > >*There exist temporal graphs containing nodes $u, v$ that have non-isomorphic TCTs, yet no TGAT nor TGN-Att with mean message aggregator (i.e., using $\mathrm{Mean}$ as $\mathrm{MsgAgg}$) can distinguish $u$ and $v$.*
> > >
> > > We would like to re-emphasize the generality of the remaining results, i.e., they are agnostic to specific message aggregators, and rely only on the fact that memory state $s_u$ is a function of events within the monotone TCT of $u$.
> > >
> > >
> > > **Delineating the role of relative positional features from that of memory**
> > >
> > > Thanks for giving us an opportunity to clarify the role of memory unit design. We've now included a description of memory in the Background section to make this clear. The definition of memory now reads as:
> > >
> > > >Let $\mathcal{J}(v, t)$ be the set of events involving $v$ at time $t$. The state update of $v$ due to the events $\mathcal{J}(v, t)$ is
> > > $$
> > > m_v(t) = \mathrm{MsgAgg}(\\{[s_v(t^-), s_u(t^-), t-t_v, e_{vu}(t)] \mid (v, u, t) \in  \mathcal{J}(v, t)\\})
> > > $$
> > > $$
> > > s_v(t) = \mathrm{Memory}(s_v(t^{-}), m_v(t)),
> > > $$
> > > where $s_v(0)=x_v$ (initial node features), $s_{v}(t^{-})$ denotes the state of $v$ right before time $t$, and $t_{v}$ denotes the time of the last update to $v$.
> > >
> > > We would also like to point out that our PE cannot be simply seen as part of the memory module. First, PE varies depending on which link we want to predict. Second, computing our positional features requires identifying a node at different levels of another node's TCT, i.e., it would require a (non-anonymous) memory that has access to node ids. Overall, if the memory is based only on local features and structural information, then MP-TGNNs are restricted. Positional encoded features, however, provide information that is not local and hence helps alleviate some of the restrictions of MP-TGNNs. We will include a discussion on this as well.
> > >
> > > Thanks, again, for your constructive feedback. Since all your concerns and questions have been adequately addressed, we hope the same would translate into a raised score.

---

> > > ### Author Response · Authors · 2022-08-09
> > > **Comparison against TGN-CAW**
> > >
> > > Thanks again for your detailed feedback. We have now uploaded a revised version of the manuscript to reflect our steps to address all your concerns. In addition, we conducted experiments to assess the performance of TGN-CAW [1] in our setting (same as the original TGN paper) as described below.
> > >
> > > Specifically, we've been able to conduct experiments on UCI and Wikipedia with model selection over the hyper-parameters n-layers $\in \\{1, 2\\}$, number of TGN neighbors $\in \\{10, 20\\}$, and caw-neighbors (walks) $\in \\{ [8, 2], [32, 1], [64, 1]\\}$. The results thus obtained are similar to those for TGN-Att --- e.g. Wikipedia $\approx$ 97% and UCI $\approx$ 77\% in the transductive setting, albeit the TGN-CAW model beats TGN-Att on UCI in the inductive setting: 74.70 (TGN-Att) vs. 77.52 (TGN-CAW) (using 2 layers, 10 temporal neighbors, and 32 2-sized walks). This is significantly worse than the values we obtain with PINT and CAW. Also, it is not clear if TGN-CAW is competitive against CAW as [1] does not include CAW in their experiments.
> > >
> > > We would also like to point out the computational issues: a single epoch of TGN-CAW (64 2-sized walks, 20 temporal neighbors, and 2 layers on Enron) can take over 12 hours on a machine with a V100 GPU and 32GB of RAM. We believe that a fairer assessment of TGN-CAW in our setup would require an extensive hyperparameter search. For instance, based on previous experiments, the optimal walk length for CAW may vary from 2 (UCI) to 5 (Enron). Such an exhaustive search would only be possible by the final version due to the time constraints of the author-reviewer discussion period.
> > >
> > > Thank you for suggesting comparisons with TGN-CAW [1]. These additional results further underscore the relative effectiveness of PINT. TGN-CAW is certainly a relevant related work, in the context of walk-aggregation based methods, so we'll be sure to suitably position its contributions in the final version.

---

> ### Author Response · Authors · 2022-08-02
> **Response to Reviewer HPYE (Part 2/2)**
>
> **Q9: "What is the square root term in equation (4)?"**
>
> Thanks for bringing this up. The root square term is an artifact we considered in the initial draft of the paper for normalization purposes. In our experiments, since we use a fixed number of sampled neighbors, this term has no effect. We will remove it in the revised version of the paper.
>
> **Q10: "It will be better to add more space between P and r in equation (9) and (10)"**
>
> Thanks for your suggestion. We will add a space to separate $P$ and $r$ in Eqs. (9) and (10).
>
> **Q11: "There is a highly related work CAW-TGN [1] that use CAW to get positional info for TGN which is similar to your paper, and you should add this into your comparison baseline"**
>
> Thanks for the pointer. Since our evaluation setup (same as TGN and TGAT) and [1] are significantly different, we can not directly compare our results against those in [1]. Thus, given the high computational cost and the lack of optimal hyper-parameter for reproducibility of [1], we are still currently running experiments regarding this comparison and will do our best to provide the results by the end of the discussion period here.
>
> **Q12: "Why some datasets do not have static method performance?"**
>
> The original TGN repo does not provide info for reproducing the results for static GNNs. Therefore, we decided to only report numbers for the datasets available in the original TGN paper. In addition, static methods are expected to perform poorly on unattributed networks (no edge features), where only temporal information is available.

---

### Official Review · Reviewer_pA52 · 2022-07-09

**Rating:** 7
**Confidence:** 3
**Soundness:** 4 excellent
**Presentation:** 3 good
**Contribution:** 4 excellent

**Summary:**

The paper provides a theoretical framework to analyse the expressive power of temporal graph networks (TGNs), and designs a novel model, PINT, which is provably more expressive than existing ones. Specifically, the paper proves the role of memory, benefits of injective message passing, limits of existing TGN models, temporal extension of the 1-WL test and its implications, impossibility results about temporal graph properties, and the relationship between main classes of TGNs. Then, PINT is introduced, with the elaborately design of injective temporal aggregation and relative positional features. Empirical investigations show that PINT is competitive or better than existing models on several real benchmarks for dynamic link prediction in different settings.

**Questions:**

1. The model does not outperform certain baselines on Enron in the transductive setting, and is relatively worse in the inductive setting. However, the authors do not provide a plausible explanation on these phenomena.

2. Several experiments are missing. For example, the results on node classification is more straighforward to show the models' power of distinguishing nodes than dynamic link prediction. Also, parameter sensitivity can be analysed on \alpha and \beta, because it may be important for controling the learned injective function.

3. Does PINT still fail in differing properties mentioned in Proposition 7, even the relative positional features are added?

4. There are some places that can be improved to make the discussion clearer. For example, I am a bit confused when reading line 45-50, because I fail to recognize the concept "injective MP-TGN", as it is referred to as "MP-TGNs equipped with injective functions" and "these most powerful MP-TGNs", and is contained in "MP-TGN" mentioned in the last sentence of this paragraph. Besides, there is no clear definition of the concept "anonymous TGNs", which comes up for several times in the paper.

5. A minor flaw: the term "discrete-time dynamic graph" in line 83 should be in italic type, aligning with line 85.

**Limitations:**

This work do not show any potential negative societal impact. The authors illustrate cases when the model fails, which is possibly related to temporal 1-WL test. Future works can be focused on going beyond this.

**Strengths And Weaknesses:**

Strengths:
1. The paper puts forward an interesting research direction of theoretically analysing the expressive power of temporal graph networks, which is of great importance yet long be ignored.
2. The theoretical results in the paper are solid. The authors extend the analysis framework in static GNNs to TGNs. The lemmas and propositions are closely linked to each other. The proofs seem logically rigorous, covering both special and general cases.
3. The design of PINT is concise yet with firm theoretical foundation.
4. Impossibility results of PINT are also introduced.

Weakness:
1. Discussion on the experimental results are not sufficient. Notice that the model does not outperform certain baselines on Enron in the transductive setting, and is relatively worse in the inductive setting. However, the authors do not provide a plausible explanation on these phenomena.
2. Several important experiments are missing. For example, the results on node classification is more straighforward to show the models' power of distinguishing nodes than dynamic link prediction.

---

> ### Author Response · Authors · 2022-08-02
> **Response to Reviewer pA52**
>
> Thank you for your comments. You have raised good points, and we have run additional experiments to address your concerns. We will update the manuscript to include these results and provide additional clarifications. We hope our answers have improved your view of the manuscript and your confidence in your assessment.
>
> **W1/Q1: "The model does not outperform certain baselines on Enron in the transductive setting, and is relatively worse in the inductive setting."**
>
> In Table 1, we show results for $(d=4)$-dimensional positional features. That being said, we believe a more extensive hyper-parameter search could yield even better results.
> As an example, Appendix F (Figure S7) provides complementary experiments showing that, by increasing the dimension of positional features to $d=10$, PINT  beats all baselines on Enron in the transductive setting (92.696 AP) and ranks second in the inductive case (88.342 AP).
>
> **W2/Q2: "results on node classification...Also, parameter sensitivity can be analysed on $\alpha$ and $\beta$"**
>
> As shown in the link prediction experiments, PINT shines in unattributed datasets (without edge features), where leveraging graph structure is paramount. Notably, both benchmarks available (Reddit and Wikipedia) for node classification rely on rich edge features. Nonetheless, the table below shows that PINT ranks first on Reddit and second on Wikipedia.
>
> The values for PINT below are based on three repetitions. We will increase the number of repetitions and include the final values in the revised manuscript, including implementation details and hyperparameters.
>
> |Node classification: |Wikipedia|Reddit|
> |-|-|-|
> | CTDNE | 75.89 | 59.43 |
> | JODIE | 84.84 | 61.83 |
> | TGAT | 83.69 | 65.56 |
> | DyRep | 84.59 | 62.91 |
> | TGN-Att | 87.81 | 67.06 |
> | PINT | 87.04 | 67.64|
>
> Regarding the choice of $(\alpha, \beta)$, note that our aggregation function (Eq. 8) can be written in terms of $c = \alpha^\beta$. In turn, we can obtain any value of $c$ by fixing $a$ and varying $\beta$. That being said, the table below shows the performance of PINT in the inductive setting for UCI and Wikipedia, with $\alpha=2$ and $\beta$ ranging from $10^{-3}$ to $10^{-5}$. Notably, the choice of $\beta$ can play an important role in performance. For instance, the performance of PINT jumps from $92.18$ to $96.01$ (on UCI) when we increase $\beta$.
>
> |Varying $\beta$: |UCI|Wikipedia|
> |-|-|-|
> |PINT w/ $\beta=10^{-3}$| 92.18 | 98.73 |
> |PINT w/ $\beta=10^{-4}$| 95.12 | 98.78 |
> |PINT w/ $\beta=10^{-5}$| 96.01 | 98.56 |
>
> **Q3: "Does PINT still fail in differing properties mentioned in Proposition 7, even the relative positional features are added?"**
>
> This is an interesting question. In fact, PINT can distinguish the constructions we provide in Proposition 7 (Figure 3). To see this, it suffices to note that the TCTs for, e.g., $u_1$ and $u_1^\prime$ are no longer isomorphic when positional features are added. Consider a 2-layer model, then $r_{ w_1 \rightarrow u_1} = [0, 1, 1]$ while $r_{w^{\prime}_1 \rightarrow u^{\prime}_1} = [0, 1, 0]$, with $w_1$ and $w_1^\prime$ as denoted in Figure 3.
>
> **Q4: "There are some places that can be improved to make the discussion clearer. For example, I am a bit confused when reading line 45-50, because I fail to recognize the concept "injective MP-TGN"...Besides, there is no clear definition of the concept "anonymous TGNs""**
>
> Thanks for pointing this out. We will improve our revised manuscript to make these excerpts clearer. By ``anonymous TGNs'', we mean TGNs that do not rely on node identifiers [1]. We will clarify it in the text, and we will be more explicit whenever possible.
>
> [1] https://openreview.net/pdf?id=B1l2bp4YwS
>
> **Q5: "A minor flaw: the term "discrete-time dynamic graph" in line 83 should be in italic type, aligning with line 85."**
>
> Thanks for the suggestion. We will update it in the revised paper.

---

> > ### Comment · Reviewer_pA52 · 2022-08-09
> > **Thanks for your response**
> >
> > Thanks for your efforts to improve the paper, which have solved most of my concerns. I would like to keep the score as is.

---

> > > ### Author Response · Authors · 2022-08-09
> > > **RE: Thanks for your response**
> > >
> > > Thanks! We appreciate your reply and support for our work.

---

### Official Review · Reviewer_gAwT · 2022-07-10

**Rating:** 8
**Confidence:** 3
**Soundness:** 4 excellent
**Presentation:** 3 good
**Contribution:** 4 excellent

**Summary:**

The paper studies the topic of graph neural networks in temporal settings, which is a very significant avenue of graph learning applications. In doing so, a number of recent prominent works in this sub-topic is converged in this paper through solid theoretical analyses. Particularly, theoretical foundations have been developed (summarized in Fig 1 with references to relevant statements) which improves the understanding of temporal GNNs and provides a basis for further improvement targeting this research domain. Standing on such theoretical studies, the paper proposes a position-encoding temporal graph network (PINT) which is provably powerful than existing GNNs for temporal graph learning, which is empirically powerful than existing methods on the benchmarks used.

**Questions:**

-What would be the effect of directly adopting positional encodings that have been developed for static GNNs (which the paper reviews in the supplementary sections), say for different static graph snapshots at different timestamps?
-In line 347-349, it is claimed that MP-TGNs are outperformed by PINT w/o pos. feat. However, from Table 1, it is observed that in 7/12 results (12 = 6 transductive + 6 inductive), it is not the case.
-With respect to the time complexity analysis, the Figure 6 has an interesting mark at epoch 25, after which PINT's runtime is faster than CAW. To this end, will it possible to show some sample evaluations of PINT at only 25 epochs (or lesser) of training? This could provide a useful insight on comparison of PINT with CAW at a situation where PINT's runtime is not better than CAW.

**Limitations:**

Yes, the limitations are clearly discussed as well as societal impact mentioned.

**Strengths And Weaknesses:**

=>Strengths:
-This work is an excellent convergence of the recent advances in extending GNNs on temporal graphs. It includes several studies in terms of theoretical property of existing temporal GNNs, what are their limitations, extending WL test analysis on temporal graphs, and ways to improve TGNs beyond MP-TGNs.
-In terms of originality, I feel the paper is greatly motivated from existing theoretical advancements of GNNs applied to static graphs (in terms of 1-WL like quantification, limitations and positional encodings). Nevertheless, this extension and presentation of the theoretical insights to temporal graph settings is novel.
-In terms of significance, the results and insights from this paper would definitely be used in further developments.
-The paper writing is clear, the related literature and existing concerns are discussed.
-Overall this paper does a fundamental contribution for the topic of developing machine learning foundations on temporal and time dynamic graphs.

=>Weaknesses:
While there seems no significant weakness in the contribution of this work, below are some minor observations:
-Are the experimental results significant looking at the score differences of PINT with best model of the literature. For instance the differences are in the ranges ~0.5 scores of AP for many results in Table 1?
-Whether positional encoding (pe) is main contributor of good performance of PINT? Would incorporating the 'pe' in existing TGNs would be at par performance of PINT+pe?

---

> ### Author Response · Authors · 2022-08-02
> **Response to Reviewer gAwT**
>
> We greatly thank you for the helpful and insightful comments. To support our discussion, we have run additional experiments --- which we will add to the supplementary material of the revised paper. We hope our answers address your concerns.
>
> **W1: "Are the experimental results significant looking at the score differences of PINT with best model of the literature. For instance the differences are in the ranges ~0.5 scores of AP for many results in Table 1?"**
>
> We note that taking into account standard deviation is important to assess statistical significance. While mean AP differences between PINT and the second-best model do not seem high on attributed networks (Twitter/Reddit/Twitter), we highlight that the corresponding standard deviations are much smaller. For instance, in table 1 (transductive setting), on 4 out of 5 datasets, the mean AP of PINT is over 5 standard deviations away from the second-best model.
>
> **W2: "Would incorporating the 'pe' in existing TGNs would be at par performance of PINT+pe?"**
>
> Both the injective aggregation and the positional encodings are important for the performance of PINT.
> PE plays a major role in datasets without edge features (UCI, Enron, and LastFM).
> For instance, Table 1 shows that PINT's performance drops from 88.06 to 81.35 (transductive) and from 91.76 to 88.44 (inductive). As suggested by the reviewer, it is also possible to incorporate our PE into existing TGNs.
> As a proof of concept, we have implemented TGN-Att with PE for the three unattributed datasets. The tables below show that TGN-Att receives a significant boost from our PE. However, PINT still beats TGN-Att+PE on 5 out of 6 cases. We will include these additional results in the revised manuscript.
>
> |Transductive: |UCI|Enron|LastFM|
> |-|-|-|-|
> |TGN-Att| 80.40 | 79.91 | 80.69 |
> |TGN-Att + PE| 95.64 | 85.04 | 89.41 |
> |PINT| 96.01 | 88.71 | 88.06|
>
>
> |Inductive: |UCI|Enron|LastFM|
> |-|-|-|-|
> |TGN-Att| 74.70 | 78.96 | 84.66 |
> |TGN-Att + PE| 92.82 | 76.27 | 91.63 |
> |PINT | 93.97 | 81.05 | 91.76 |
>
> **Q1: "What would be the effect of directly adopting positional encodings that have been developed for static GNNs?"**
>
> Thanks for the question. We think leveraging static PE in dynamic graph tasks is a promising direction. For the discrete-time case, using them might be straightforward: apply PE to augment node features in each snapshot and use a recurrent model to combine temporal embeddings. For continuous time (PINT's setting), we might define snapshots based on timestamps. The main challenge here would be to decide the granularity of snapshots. For instance, using singleton timestamps might yield very sparse snapshots and skyrocket computational costs. On the other hand, having large time windows could eclipse fine-grained graph dynamics.
>
> **Q2: "it is claimed that MP-TGNs are outperformed by PINT w/o pos. feat. However ... it is not the case."**
>
> Thanks for pointing this out. The sentence should convey that "MP-TGNs are outperformed by PINT w/o pos in datasets without edge features". We have corrected this in the revised version of the paper.
>
> **Q3: "will it possible to show some sample evaluations of PINT at only 25 epochs (or lesser) of training?"**
>
> Below, we report the performance of PINT with max 5,10, and 15 epochs. Overall, PINT is still improving but already better than CAW.
>
> |Transductive performance: |UCI|Wikipedia|
> |-|-|-|
> | PINT w/ <=5 epochs | 95.70  +- 0.10 | 98.68 +- 0.04  |
> | PINT w/ <=10 epochs | 95.94 +- 0.17 | 98.76 +- 0.04 |
> | PINT w/ <=15 epochs | 95.91 +- 0.23 | 98.73 +- 0.07 |
>
>
> |Inductive performance: |UCI|Wikipedia|
> |-|-|-|
> | PINT w/ <=5 epochs | 93.34  +- 0.14 | 98.22 +- 0.02 |
> | PINT w/ <=10 epochs | 93.70 +- 0.25 | 98.34 +- 0.06 |
> | PINT w/ <=15 epochs | 93.74 +- 0.28 | 98.35 +- 0.05 |

---

> > ### Comment · Reviewer_gAwT · 2022-08-08
> > **Response to Authors**
> >
> > Thank you for your answers for addressing the minor observations that I raised. Also, thanks for running additional experiments in this short period of time to support the answers. It confirms that 'pe' is a major contributor of the performance looking at TGN-Att results, at least for the cases with unattributed datasets. I have no other concerns and keep the accept score as is.

---

> > > ### Author Response · Authors · 2022-08-09
> > > **RE: Response to Authors**
> > >
> > > Many thanks, again, for your feedback and engagement in the discussion. We are grateful for your strong support for this work.

---

### Official Review · Reviewer_9vm8 · 2022-07-11

**Rating:** 7
**Confidence:** 3
**Soundness:** 3 good
**Presentation:** 3 good
**Contribution:** 3 good

**Summary:**

The paper attempts to address the limitations of message-passing TGNs (MP-TGNs) and walk-aggregating TGNs (WA-TGNs). To be specific, a so-called position-encoding injective temporal graph net (PINT) is proposed, which not only defines injective message passing and update steps, but augments memory states with novel relative positional features. Note that, the time complexity of computing and communicating positional features in PINT is less than CAW. Experimental results show that PINT outperforms existing TGNs on several real-world benchmarks in both transductive and Inductive settings.

**Questions:**

Questions:
1. Can the proposed PINT deal with long-term dependencies?
2. For edge and node embeddings, why the authors consider 10/20-hop neighbors? Will the small number of neighbors will impact the performance?
3. I am curious whether the proposed PINT model can be used for node classification/forecasting?
4. The standard deviations of PINT on Reddit and Twitter (for both transductive and Inductive settings) are very small. I wonder is there any reasons/ways to interpret this scenario?
5. For Eqs. 9 and 10. Why the authors only do multiplication between $P$ and $r^{t^-}_{i \rightarrow u}$ (e.g., in Eq. 9)?

**Limitations:**

I have no ethical concerns with this paper.

**Strengths And Weaknesses:**

Strengths:
1. The paper is clearly written and easy to follow. The Introduction and Related Works (Preliminaries) parts are well written.
2. Visualizations and plots are intuitive.
3. Building injective aggregation and update functions in the temporal setting is interesting and somewhat novel.
4. Experiments are extensive and promissing, considering both transductive and Inductive settings.

Weaknesses:
Although the authors have already show that PINT can be trained faster than CAW (which is the main competitor shown in Table 1). I wonder will the large-scale temporal networks limit the behavior of PINT?

---

> ### Author Response · Authors · 2022-08-02
> **Response to Reviewer 9vm8**
>
> Thank you for your valuable feedback. Based on your review, we will update our paper to show results on node classification, add comments regarding small variance, and clarify the intuition behind Eq. (9). We hope our answers have sufficiently addressed your concerns, and if that is the case, we kindly ask you to consider increasing your score.
>
> **"I wonder will the large-scale temporal networks limit the behavior of PINT?"**
>
> As shown in Figures 6 and S6, PINT's computational overhead can be amortized during training. This implies that we only need to compute updates for the first epoch and makes PINT much faster than the previous SOTA (i.e., CAW). That being said, we believe our work could inspire follow-ups focusing on faster approximation methods, e.g., using Nystrom's approximation or random features [1].
>
> [1] https://arxiv.org/abs/2009.14794
>
> **Q1: "Can the proposed PINT deal with long-term dependencies?"**
>
> Since PINT employs LSTM-based memory, in principle, it allows nodes to store long-range historical information --- without the computational burden of increasing the number of message-passing layers. Nonetheless, we believe that evaluating whether or how fast temporal GNNs forget events (in practice) is an important avenue for future works.
>
> **Q2: "For edge and node embeddings, why the authors consider 10/20-hop neighbors? Will the small number of neighbors will impact the performance?"**
>
> We note that 10/20 refers to the number of 1-hop neighbors, not the neighborhood depth. The main reason why people restrict neighborhood size is that it determines the branching factor of TCTs, i.e., it directly impacts the computational cost. Notably, these numbers are also used by TGN/TGAT, and we do the same to ensure a fair comparison.
>
> **Q3: "I am curious whether the proposed PINT model can be used for node classification/forecasting?"**
>
> We can extract node embeddings as described in the paragraph *edge and node embeddings* (line 278 in Section 4). Subsequently, a classifier (MLP) can be used to make a prediction for the node. As shown in the link prediction experiments, PINT shines in unattributed datasets (without edge features), where leveraging graph structure is paramount. Notably, both benchmarks available (Reddit and Wikipedia) for node classification rely on rich edge features. Nonetheless, the table below shows that PINT ranks first on Reddit and second on Wikipedia.
>
> The values for PINT below are based on three repetitions. We will increase the number of repetitions and include the final values in the revised manuscript, including implementation details and hyperparameters.
>
>
> |Node classification: |Wikipedia|Reddit|
> |-|-|-|
> | CTDNE | 75.89 | 59.43 |
> | JODIE | 84.84 | 61.83 |
> | TGAT | 83.69 | 65.56 |
> | DyRep | 84.59 | 62.91 |
> | TGN-Att | 87.81 | 67.06 |
> | PINT | 87.04 | 67.64|
>
> **Q4: "The standard deviations of PINT on Reddit and Twitter (for both transductive and Inductive settings) are very small. I wonder is there any reasons/ways to interpret this scenario?"**
>
> Thanks for pointing this out. In fact, in the majority of cases PINT achieves significantly smaller standard deviations. The same does not occur when we remove our positional features. For instance, in Reddit and Twitter, the standard deviations increase $\geq \times 4$. For some cases, we observe these numbers still drop if we increase the dimension of the positional encoding (see, e.g., results for Enron with $d=10$ in Appendix F). This might be evidence that positional encodings are a strong (and useful) inductive bias for continuous-time TGNs.
>
> **Q5: "For Eqs. 9 and 10. Why the authors only do multiplication between $P$ and $r_{i \rightarrow u}^{t^-}$ (e.g., in Eq. 9)?"**
>
> Thanks for the opportunity to clarify the update in Eq. (9). In this regard, Figure S5 (Appendix) provides a good illustration of the intuition behind Eq. (9).
>
> Let us consider an event $(u, v, t)$.
> This event modifies the TCT of $v$ by appending the TCT rooted at $u$ as a child of the node $v$ (i.e., $u$ becomes an additional child of $v$). Therefore, to update the counts of a given node $i$ in the TCT of $v$, we simply need to add to the previous counts the occurrences of $i$ from the TCT of $u$, i.e., $r^{(t^-)}_{i \rightarrow u}$.
>
> To account for the fact that added elements are attached to layer 1 (instead of 0), we apply the shifting matrix $P$ to  $r^{(t^-)}_{i \rightarrow u}$. This leads to the update in Eq. (9).

---

> > ### Comment · Reviewer_9vm8 · 2022-08-09
> > **Thanks for the response!**
> >
> > Thank you for the response! Most of my concerns have been addressed. I would like to increase my score by one point (from 6 to 7).

---

> > > ### Author Response · Authors · 2022-08-09
> > > **RE: Thanks for the response!**
> > >
> > > Thanks again! We sincerely appreciate your input and engagement in the discussion.

---

### Author Response · Authors · 2022-08-09
**Thanks for your service**

We are grateful to all the reviewers for their time and insightful comments, as well as to the (senior) area, program, and general chairs for their service to the community.

We are glad to note the positive response of all the reviewers, and specifically, their acknowledgments that our work consists of a fundamental contribution for ML on temporal graphs (gAwT) with extensive experiments (9vm8) that builds on concrete failure cases of SOTA methods (HPYE), and is backed by solid theoretical results (pA52). We also appreciate that the reviewers find the work well-written (9vm8, gAwT) and novel (9vm8, gAwT) with potential impact on further developments (pA52) and new benchmarks (HPYE).

To the best of our efforts, we’ve diligently tried to address all the specific comments, including the minor ones, that have been raised by each reviewer. We’ve also revised the paper in the meantime to reflect the same (changes are indicated in blue). In particular, some of the main revisions are:
1. Additional experimental results on node classification (Table S5);
2. Adjustment to the statement of Prop. 1;
3. Extension of Prop. 7 to include CAW;
4. Clarifications regarding the assumption of mean-based aggregation (made explicit now)  in Prop. 4, as well as the memory.
5. Additional results regarding TGN-Att with positional features (Table S4);
6. Exposition of the intuition underlying positional features (Figure 4);

We believe that acting on reviewers’ feedback has reinforced the many strengths of this work, and we thank them again for their very constructive comments.

---

### Meta-Review · Area_Chair_5CbE · 2022-08-23

**Recommendation:** Accept
**Confidence:** Certain

**Metareview:**

This paper considered temporal graph networks (TGNs). It first analyzed the representational power and limits of the two main categories of TGNs (WA-TGN and MP-TGN), proving neither category subsumes the other. It extended the 1-WL (Weisfeiler-Leman) test to TGNs and showed when TGNs become as expressive as the temporal WL. It also showed that sufficiently deep MP-TGNs cannot benefit from memory, and MP-TGNs fail to compute graph properties such as girth. Based on the theoretical results, it proposed a provably more expressive TGN called PINT and showed that it outperforms existing methods on several real-world benchmarks.

The work has made solid and novel contributions, including theoretical studies of the expressive power of TGNs, a new framework, and strong empirical performance of the proposed framework. The authors also addressed well the comments from the reviewers and further strengthened the work during the response period.

**Award:**

No

---

### Decision · Program_Chairs · 2022-09-14

Accept